# Manipulation of the unfolded protein response: A pharmacological strategy against coronavirus infection

Liliana Echavarría-Consuegra[1☯], Georgia M. Cook[1☯], Idoia Busnadiego[2], Charlotte Lefèvre[1], Sarah Keep[3], Katherine Brown[1], Nicole Doyle[3], Giulia Dowgier[3], Krzysztof Franaszek[1], Nathan A. Moore[4¤], Stuart G. Siddell[4], Erica Bickerton[3], Benjamin G. Hale[2], Andrew E. Firth[1], Ian Brierley[1], Nerea Irigoyen[1]*

**1** Division of Virology, Department of Pathology, University of Cambridge, Tennis Court Road, Cambridge, United Kingdom, **2** Institute of Medical Virology, University of Zurich, Zurich, Switzerland, **3** The Pirbright Institute, Woking, Surrey, United Kingdom, **4** Department of Cellular and Molecular Medicine, University of Bristol, Bristol, United Kingdom

☯ These authors contributed equally to this work.
¤ Current address: Basingstoke and North Hampshire Hospital, Hampshire Hospitals, NHS Foundation Trust, Basingstoke, United Kingdom.
* ni236@cam.ac.uk

**Data Availability Statement:** RiboSeq and RNASeq sequencing data have been deposited in

## Abstract

Coronavirus infection induces the unfolded protein response (UPR), a cellular signalling pathway composed of three branches, triggered by unfolded proteins in the endoplasmic reticulum (ER) due to high ER load. We have used RNA sequencing and ribosome profiling to investigate holistically the transcriptional and translational response to cellular infection by murine hepatitis virus (MHV), often used as a model for the *Betacoronavirus* genus to which the recently emerged SARS-CoV-2 also belongs. We found the UPR to be amongst the most significantly up-regulated pathways in response to MHV infection. To confirm and extend these observations, we show experimentally the induction of all three branches of the UPR in both MHV- and SARS-CoV-2-infected cells. Over-expression of the SARS-CoV-2 ORF8 or S proteins alone is itself sufficient to induce the UPR. Remarkably, pharmacological inhibition of the UPR greatly reduced the replication of both MHV and SARS-CoV-2, revealing the importance of this pathway for successful coronavirus replication. This was particularly striking when both IRE1α and ATF6 branches of the UPR were inhibited, reducing SARS-CoV-2 virion release (~1,000-fold). Together, these data highlight the UPR as a promising antiviral target to combat coronavirus infection.

## Author summary

SARS-CoV-2 is the novel coronavirus responsible for the COVID-19 pandemic which has resulted in over 150 million cases since the end of 2019. Most people infected with the virus will experience mild to moderate respiratory illness and recover without any special treatment. However, older people, and those with underlying medical problems like

the ArrayExpress database (http://www.ebi.ac.uk/arrayexpress) under the accession numbers E-MTAB-8650 and E-MTAB-8651.

**Funding:** GMC: Wellcome Trust Four-year PhD Studentship (203864/Z/16/Z) https://wellcome.org/grant-funding/schemes/four-year-phd-programmes-studentships-basic-scientists EB: Biotechnology and Biological Sciences Research Council (BBSRC. UKRI): BBS/E/I/00007034 and BBS/E/I/00007031 grants. https://bbsrc.ukri.org/funding/ BGH: Swiss National Science Foundation grant (31003A_182464) http://www.snf.ch/en/funding/Pages/default.aspx AEF: Wellcome Trust grant (106207, https://wellcome.org/grant-funding) and a European Research Council grant (646891, https://erc.europa.eu/) IBr: Medical Research Council (MRC. UKRI) Project Grant (MR/M011747/1, https://mrc.ukri.org/funding/how-we-fund-research/research-grant/) and a Wellcome Trust Investigator Award (202797/Z/16/Z, https://wellcome.org/grant-funding/schemes/investigator-awards-science). NI: Isaac Newton Trust Grant (18.40r, https://www.newtontrust.cam.ac.uk/ResearchGrants), a Royal Society Research Grant (RGS\R1\191137, https://royalsociety.org/grants-schemes-awards/grants/research-grants/) and an Isaac Newton Trust/Wellcome Trust ISSF/University of Cambridge Joint Research Grant (https://www.research-strategy.admin.cam.ac.uk/research-funding/wellcome-trust-institutional-strategic-support-fund-issf/funding-calls/joint). The funders had no role in study design, data collection and analysis, decision to publish, or preparation of the manuscript.

**Competing interests:** The authors have declared that not competing interests exist.

chronic respiratory disease are more likely to develop a serious illness. So far, more than 3 million people have died of COVID-19. Unfortunately, there is no specific medication for this viral disease.

In order to produce viral proteins and to replicate their genetic information, all coronaviruses use a cellular structure known as the endoplasmic reticulum or ER. However, the massive production and modification of viral proteins stresses the ER and this activates a compensatory cellular response that tries to reduce ER protein levels. This is termed the unfolded protein response or UPR. We believe that coronaviruses take advantage of the activation of the UPR to enhance their replication.

The UPR is also activated in some types of cancer and neurodegenerative disorders and UPR inhibitor drugs have been developed to tackle these diseases. Here, we show also that these compounds can significantly reduce SARS-CoV-2 replication in human lung cells.

## Introduction

The *Coronaviridae* are a family of enveloped viruses with positive-sense, non-segmented, single-stranded RNA genomes. Coronaviruses (CoVs) cause a broad range of diseases in animals and humans. SARS-CoV, MERS-CoV and SARS-CoV-2, members of the genus *Betacoronavirus*, are three CoVs of particular medical importance due to high mortality rates and pandemic capacity [1–3]. SARS-CoV-2 is the causative agent of the current COVID-19 pandemic, which has resulted in over 150 million cases and more than 3 million deaths since the end of 2019. Although up to 15% of the cases develop a severe pathology [4,5], no specific therapeutic treatment for COVID-19 has been approved to date, highlighting the urgent need to identify new antiviral strategies to combat SARS-CoV-2, besides future CoV zoonoses.

During CoV replication, the massive production and modification of viral proteins, as well as virion budding-related endoplasmic reticulum (ER) membrane depletion, can lead to overloading of the folding capacity of the ER and consequently, ER stress [6]. This activates the unfolded protein response (UPR) which is controlled by three ER-resident transmembrane sensors: inositol-requiring enzyme-1 α (IRE1α), activating transcription factor-6 (ATF6), and PKR-like ER kinase (PERK), each triggering a different branch of the UPR (Fig 1A). Activation of these pathways leads to decreased protein synthesis and increased ER folding capacity, returning the cell to homeostasis [7].

Here, we characterise global changes in the host translatome and transcriptome during murine coronavirus (MHV) infection using RNA sequencing (RNASeq) and ribosome profiling (RiboSeq). MHV is a member of the *Betacoronavirus* genus and is widely used as a model to study the replication and biology of members of the genus. In this analysis, the UPR is one of the most significantly enriched pathways. We further confirm the activation of all three branches of the UPR in MHV-infected cells. Extending our investigation to SARS-CoV-2, we find that infection with this novel CoV also activates all three UPR pathways. Moreover, we demonstrate that individual over-expression of SARS-CoV-2 ORF8 and S proteins is sufficient to induce the UPR. Remarkably, pharmacological inhibition of the UPR had a dramatic negative effect on MHV and SARS-CoV-2 replication, suggesting that CoVs may subvert the UPR to their own advantage. These results reveal that pharmacological manipulation of the UPR can be used as a therapeutic strategy against coronavirus infection.

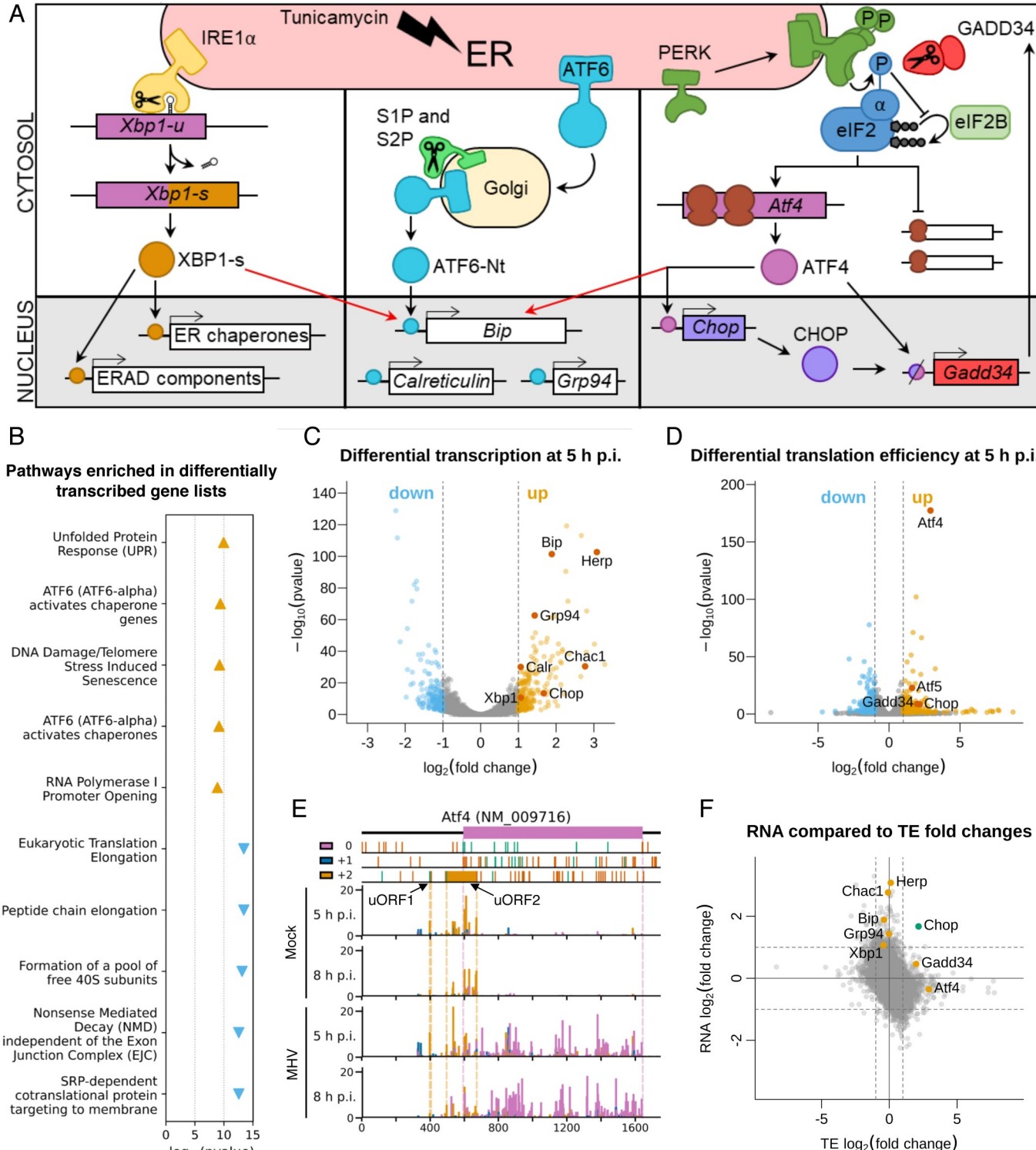

**Fig 1. Ribosome profiling reveals the unfolded protein response as a key pathway in the host response to MHV-A59 infection. (A)** Schematic of the three branches of the UPR (IRE1α, ATF6, and PERK). ERAD = ER-associated protein degradation. **(B)** Top five most significantly enriched Reactome pathways [11] associated with the lists of transcriptionally up-regulated genes (orange triangles pointing upwards) and transcriptionally down-regulated genes (blue triangles pointing downwards), plotted according to the false discovery rate (FDR)-corrected *p* value of the enrichment. Full results, including pathway IDs, are in S3 Table. **(C)** Volcano plot showing the relative change in abundance of cellular transcripts and the FDR-corrected *p* value for differential expression between the mock and infected samples (n = 2 biological replicates). Grey vertical lines indicate a transcript abundance fold change of 2. Genes which have fold changes

greater than this threshold and a $p \leq 0.05$ value of less than 0.05 are considered significantly differentially expressed and coloured orange if up-regulated and blue if down-regulated. Selected genes are annotated in red. **(D)** Volcano plot showing the relative change in translation efficiency of cellular transcripts, and the FDR-corrected $p$ value, between the mock and infected samples (n = 2 biological replicates). Colours and fold change and $p$ value thresholds as in C. **(E)** Analysis of RPFs mapping to *Atf4* (NCBI RefSeq mRNA NM_009716). Cells were infected with MHV-A59 or mock-infected and harvested at 5 h p.i. (libraries from replicate 2) or 8 h p.i. RPFs are plotted at the inferred position of the ribosomal P site and coloured according to phase (which position within the codon the 5′end of the read maps to: pink for 0, blue for +1, yellow for +2). The main ORF (0 frame) is shown at the top in pink, with start and stop codons in all three frames marked by green and red bars (respectively) in the three panels below. The two yellow rectangles in the +2 frame indicate the known *Atf4* uORFs (the first of which is only three codons in length). Dotted lines serve as markers for the start and end of the features in their matching colour. Note that read densities are plotted as reads per million host-mRNA-mapping reads, and that bar widths were increased to 12 nt to aid visibility, and therefore overlap, and were plotted sequentially starting from the 5′ end of the transcript. **(F)** Plot of $\log_2$(fold changes) of translation efficiency (TE) vs transcript abundance for all genes included in both analyses. Grey lines indicate fold changes of 2. Fold changes are plotted without filtering for significant $p$ values. Selected genes are marked: genes up-regulated predominantly by one of either transcription or TE are marked in orange (upper middle and right middle sections), and *Chop*, which is up-regulated at the level of both transcription and TE, is marked in green (top right section).

## Results

### Differential gene expression analysis of murine cells infected with MHV-A59

To survey genome-wide changes in host transcription and translation during CoV infection, murine 17 clone 1 cells (17 Cl-1) were infected with recombinant MHV-A59 at a multiplicity of infection (MOI) of 10, or mock-infected, in duplicate and harvested at 5 hours post-infection (h p.i.). Lysates were subjected to RNASeq and parallel RiboSeq [8,9], which allows global monitoring of cellular translation by mapping the positions and abundance of translating ribosomes on the transcriptome with sub-codon precision. Quality control analysis confirmed the libraries were of high quality (S1 Fig and S1 Table).

To assess the effects of MHV infection on cellular transcript abundance, differential expression analysis was performed at 5 h p.i. with DESeq2 [10] (Fig 1B and 1C and S2 and S3 Tables). At this timepoint, viral RNA synthesis approaches a maximum, but it precedes the onset of cytopathic effects such as syncytium formation [9]. Between infected and mock-infected conditions, genes with a fold change $\geq 2$ and a false discovery rate (FDR)-corrected $p$ value of $\leq 0.05$ were considered to be significantly differentially transcribed (S2 Table). To determine the biological pathways involved in the response to infection, we carried out Reactome pathway enrichment analysis [11] on the lists of significantly differentially transcribed genes (Fig 1B and S3 Table). The most significantly enriched pathway associated with transcriptionally up-regulated genes was "Unfolded Protein Response" (R-HSA-381119, $p = 1.1 \times 10^{-10}$), and pathways denoting the three branches of the UPR (ATF6 branch: R-HSA-381183, PERK branch: R-HSA-380994, IRE1$\alpha$ branch: R-HSA-381070) were also significantly enriched (S3 Table). Consistent with this, gene ontology (GO) term enrichment analysis of the transcriptionally up-regulated gene list revealed that UPR-related GO terms, such as "response to unfolded protein" (GO:0006986), were significantly enriched (S3 Table). Many of the enriched pathways and GO terms associated with transcriptionally down-regulated genes are related to protein synthesis, again highlighting this as a key theme of the host response.

We provide the full database of differentially expressed genes and enriched pathways/GO terms for further exploration (S2 and S3 Tables) but in this manuscript we will focus predominantly on the UPR, which has been recognised as a host response to several CoVs due to the extensive dependence of CoV replication on the ER [6]. Accordingly, some of the most differentially transcribed genes are involved in the UPR, such as *Herp* (also known as *Herpud*), *Chac1*, *Bip* (also known as *Grp78* or *Hspa5*), *Chop* (also known as *Ddit3* or *Gadd153*) and *Grp94* (also known as *Hsp90b1*) (Fig 1C).

To evaluate differences at the level of translation, we calculated relative translation efficiencies (TE; defined herein as the ratio of ribosome-protected-fragment [RPF] to total RNA

density in the CDS of a given gene) at 5 h p.i. using Xtail [12], applying the same fold change and *p*-value thresholds as for the transcription analysis. As shown in Fig 1D, several of the translationally up-regulated genes encode key proteins involved in activation of the UPR, for example ATF4, ATF5 and CHOP, which are effector transcription factors [13–18]. GADD34 (also known as MYD116/PPP1R15A), a protein that acts as a negative regulator to diminish prolonged UPR activation [19,20], was also translationally up-regulated.

Given that UPR activation can lead to eIF2α phosphorylation and host translational shut-off, we investigated whether the list of mRNAs found to be preferentially translated during MHV infection was enriched for genes resistant to translational repression by phosphorylated eIF2α (p-eIF2α) (Materials and Methods and S4 Table). We found a 9.15-fold enrichment of p-eIF2α resistant genes ($p = 1.42 \times 10^{-4}$, Fisher Exact Test). Resistance to the effects of p-eIF2α has been linked to the presence of efficiently translated upstream open reading frames (uORFs) in the 5′ UTR [13–18,21]. To investigate this in our dataset, we analysed ribosome occupancy of the main ORF compared to the uORFs on *Atf4*, a well-studied example [14] (Fig 1E). Translation of the short (three codon) uORF1 was observed under all conditions. In mock-infected samples, uORF2 was efficiently translated, largely precluding translation of the main ORF (pink). In contrast, in MHV-infected cells, a large proportion of ribosomes scan past uORF2 to translate the main ORF. This is consistent with previous studies on *Atf4* transla-tion under conditions of eIF2α phosphorylation, in which many ribosomes cannot reassemble a competent initiation complex before reaching uORF2 [13,14]. This facilitates increased pro-duction of ATF4 even when translation of most mRNAs is inhibited.

Comparison of the fold changes at the transcriptional and translational level for individual cellular mRNAs provides insight into the overall effect on gene expression (Fig 1F). Genes reg-ulated in opposing directions transcriptionally and translationally likely result in a small over-all change in expression, whereas genes regulated only in one direction likely result in a greater overall change. Many UPR genes fall into the latter category (orange points, top-centre and right-centre), reflecting published knowledge about the induction of these genes specifically at the transcriptional [22–25] or translational level [14–18,21]. *Chop* (green point, upper-right) is a rare example of a gene that is significantly up-regulated both transcriptionally and transla-tionally during MHV infection. This reflects the fact that it is transcriptionally induced by ATF4 during UPR activation and translationally p-eIF2α-resistant [26,27].

Together, the ribosome profiling results highlight the UPR as a key pathway in the host response to MHV infection, with many of the greatest expression changes observed for UPR-related genes.

## MHV infection and activation of the unfolded protein response

To further explore the extent of UPR activation during MHV infection, we investigated each of the three branches individually (Fig 1A), building on the work of several groups [28–32].

**Monitoring the PERK-eIF2α-ATF4 branch.** Upon ER stress, PERK oligomerises and auto-phosphorylates [33]. Activated PERK phosphorylates the α-subunit of eIF2 which in turn impairs recycling of inactive eIF2-GDP to active eIF2-GTP, resulting in a general shutdown of protein synthesis [34]. However, translation of ATF4 is increased in this situation [13,14,35] leading to the induction of its target genes *Chop* and *Gadd34* (Fig 1A, right). To assay PERK activation, we monitored expression of PERK, CHOP, ATF4 and p-eIF2α, by qRT-PCR and western blotting. 17 Cl-1 cells were infected with MHV-A59 or incubated with tunicamycin and harvested at 2.5, 5, 8 and 10 h. Tunicamycin, used as a positive control, is a pharmacologi-cal inducer of ER stress which activates all UPR signalling pathways. From 5 h p.i. onwards in MHV-infected cells, and at all timepoints in tunicamycin-treated cells, ATF4 and p-eIF2α

were detected and multiple bands were observed for PERK (Fig 2A) corresponding to the auto-phosphorylated species, indicative of activation of this kinase upon ER stress. In addition, as shown in Fig 2B, *Chop* and *Gadd34* mRNA levels in MHV-infected cells (blue squares) increased from 2.5 to 8 h p.i., similarly to tunicamycin-treated cells (red circles), indicating their induction by the transcription factor ATF4. At 10 h p.i., the level of p-eIF2α decreases slightly compared to its peak at 8 h p.i., consistent with GADD34 stimulating its dephosphorylation.

Virus-induced inhibition of translation as a consequence of eIF2α phosphorylation was confirmed by analytical polysome profiling in 17 Cl-1 cells (Fig 2C, upper panel), revealing the accumulation of monosomes (80S) in MHV-infected cells at 5 h p.i. In higher salt profiles (400 mM KCl; Fig 2C, lower panel), where 80S ribosomes lacking mRNA dissociate into constituent subunits, a large reduction in 80S ribosomes was seen. These data are highly consistent with inhibition of translation initiation and show that the vast majority of 80S ribosomes accumulating at this time point are not mRNA-associated. These data support the view that MHV infection leads to translational shut-off via inhibited initiation, consistent with the effects of eIF2α phosphorylation.

**Monitoring the IRE1α-XBP1 branch.** Activated IRE1α (Fig 1A, left) removes a 26-nt intron from unspliced *Xbp1* (*Xbp1-u*) mRNA leading to a translational reading frame shift and a longer protein [25,36]. The product of spliced *Xbp1* mRNA (XBP1-s) is an active transcription factor that up-regulates the expression of ER-associated degradation (ERAD) components and ER chaperones. To study this, we analysed *Xbp1-u* and *Xbp1-s* mRNAs by reverse transcriptase PCR (RT-PCR), using specific primers flanking the splice site (Fig 2D). At all timepoints, *Xbp1-u* was the predominant form in mock-infected cells whereas *Xbp1-s* was the major species in tunicamycin-treated cells. In virus-infected cells, *Xbp1-s* became predominant at 5 h p.i. This was corroborated at the translational level in the ribosome profiling datasets, in which infected samples showed increased translation of the extended ORF (yellow) generated by splicing (S2 Fig). An increase in active XBP1-s transcription factor was further supported by the finding that two of its target genes are transcriptionally up-regulated in infected cells (*ERdj4*–2.44-fold increase $p = 6.63 \times 10^{-8}$; and *P58ipk*– 1.94-fold increase $p = 3.97 \times 10^{-11}$) (S2 Table). These data indicate that the IRE1α-Xbp1 pathway is activated by MHV infection.

**Monitoring the ATF6 branch.** The ATF6 branch is activated when ATF6 translocates from the ER to the Golgi apparatus, where it is cleaved [37]. After cleavage, the amino-terminus of ATF6 (ATF6-Nt) translocates to the nucleus to up-regulate ER chaperones (Fig 1A, middle). To monitor this pathway, 17 Cl-1 cells were infected with MHV-A59 or incubated with tunicamycin and analysed by western blotting (to detect ATF6 cleavage) or by immunofluorescence (to detect ATF6 nuclear translocation) (S3A, S3B and S3C Fig). However, we were unable to detect the trimmed version of ATF6 nor a clear nuclear translocation. As ATF6-Nt was also not visible in the positive control tunicamycin-treated cells, it is likely that the antibodies used do not efficiently recognise mouse ATF6-Nt in this context.

As an alternative approach, we monitored the induction of *Bip*, *Grp94* and *Calreticulin*, transcriptionally up-regulated genes in the Reactome category "ATF6 (ATF6-alpha) activates chaperone genes" (S3 Table) and known to be induced by ATF6-Nt [38,39]. BiP mRNA or protein levels are often used as a proxy for activation of the ATF6 pathway; however, its transcription can eventually be regulated by other UPR factors such as XBP1 [40] and ATF4 [41], so it can also be used as general readout of ER stress [28,38]. Cells were harvested at 2.5, 5 and 8 h p.i. and analysed by qRT-PCR (Fig 2E). An increase in *Bip* transcription was observed in tunicamycin-treated (red circles) and to a lesser extent in MHV-infected cells (blue squares) from 2.5 to 8 h p.i., whereas mock-infected cells (green triangles) showed no induction. Despite the transcriptional up-regulation and a noticeable increase in RiboSeq reads mapping

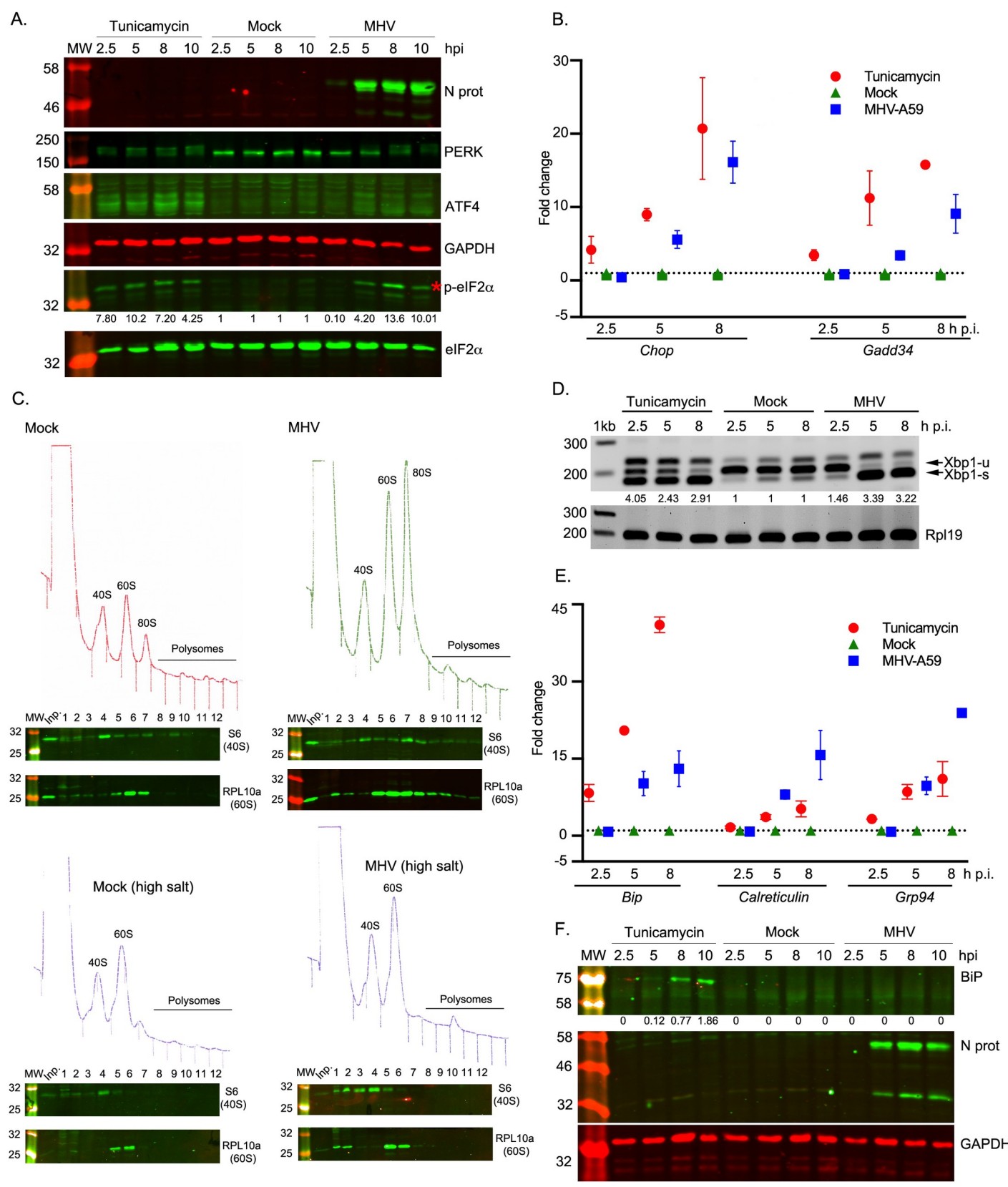

**Fig 2. MHV infection and activation of the unfolded protein response.** 17 Cl-1 cells were incubated in the presence of tunicamycin (2 μg/ml) or infected with MHV-A59 (MOI 5) and harvested at 2.5, 5, 8 and 10 h p.i. **(A)** Western blot analysis of ATF4, p-eIF2α, eIF2α, PERK and MHV N proteins. GAPDH and eIF2α were used as loading controls. Molecular masses (kDa) are indicated on the left and the p-eIF2α band is indicated by a red asterisk. Protein band quantifications for p-eIF2α, normalised by eIF2α and given relative to the timepoint-matched mock value, are provided below the immunoblot. **(B)** RT-qPCR of *Chop* and *Gadd34* mRNA for three biological replicates of a timecourse of MHV infection or tunicamycin treatment. Data are normalised using *Rpl19* as a housekeeping gene and presented as fold change of expression relative to mock-infected cells (marked as a dashed line). **(C)** Mock-infected (left upper panel) and MHV-infected (right upper panel) 17 Cl-1 cells were harvested at 5 h p.i. Cytoplasmic lysates were resolved on 10–50% sucrose density gradients. Gradients were fractionated and fractions monitored by absorbance ($A_{254}$ nm). Twelve numbered fractions were collected and proteins extracted, resolved by 12% SDS-PAGE and analysed by immunoblotting using the indicated antibodies (anti-S6 as 40S marker, anti-RPL10 as 60S marker). Mock-infected (left lower panel) and MHV-infected (right lower panel) 17 Cl-1 cells were harvested at 5 h p.i. in high-salt lysis buffer (400 mM KCl) and analysed as described above. Lane "Inp" contains whole cell lysate. **(D)** RT-PCR analysis of *Xbp1-u* and *Xbp1-s* mRNAs. *Rpl19* RT-PCR product was used as a loading control. Molecular size markers (nt) are indicated on the left. *Xbp1* splicing was quantified as the ratio *Xbp1-s* / (*Xbp1-s* + *Xbp1-u*), and the extent of splicing relative to the timepoint-matched mock is shown below each lane. The band that migrates above *Xbp1-u* is thought to represent a duplex of *Xbp1-s* and *Xbp1-u*, known as the "hybrid" band (*Xbp1-h*) [116]. **(E)** RT-qPCR of *Bip*, *Calreticulin* and *Grp94* mRNA for three biological replicates of a timecourse of MHV infection or tunicamycin treatment. Data are normalised as in B. **(F)** Cell lysates were analysed by 12% SDS-PAGE and immunoblotted using anti-BiP and anti-N antibodies. GAPDH was used as a loading control. Protein band quantifications for BiP were normalised to GAPDH. Immunoblots and agarose gels are representative of three biological replicates.

to BiP (S3D Fig), the protein was not detectable by western blot in MHV-infected cells (Fig 2F). It is not yet clear why this is the case, although down-regulation of BiP at the protein level has previously been observed during infection with other members of the order *Nidovirales* [32,42]. Alternatively, it is possible that BiP might accumulate to levels detectable by western blot at later timepoints, but the extensive cell death that occurs shortly after 10 h p.i. makes this difficult to test. Nevertheless, an increase in *Calreticulin* and *Grp94* transcription (Fig 2E) was observed in tunicamycin-treated cells (red circles) and to a greater extent in MHV-infected cells (blue squares) especially at 8 h p.i. This indicates that the ATF6 pathway is highly up-regulated during MHV-infection. Together with our studies of PERK-eIF2α-ATF4 and IRE1α-XBP1 above, these data confirm that MHV infection induces all three branches of the UPR.

## Effect of UPR inhibitors on MHV replication

Based on the strong UPR activation brought about by MHV infection, we hypothesised that pharmacological manipulation of this pathway could be used to modulate viral replication. First, we determined cell viability in 17 Cl-1 cells after drug treatment using CellTiter-Blue and trypan blue exclusion assays (S4 Fig). Subsequently, we evaluated the inhibitory effect of four different UPR inhibitors (UPRi) on each one of the UPR branches in cells infected with MHV for 8 h at MOI 5 (S5 Fig).

GSK-2606414 (henceforth referred to as PERKi) is a specific inhibitor of PERK [43,44]. As expected, PERKi treatment prevented autophosphorylation of PERK and reduced phosphorylation of its substrate, eIF2α (S5A Fig), effectively blocking this branch of the UPR. Pulse labelling of infected cells for one hour at 5 h p.i. revealed a modest increase of both viral and host protein synthesis, with no effect on mock-infected cells (S5B Fig). Analytical polysome profiling of MHV-infected cells treated with 5 μM PERKi for 5 h (S5C Fig) revealed a decrease in the accumulation of monosomes (80S) compared to MHV-infected cells at 5 h p.i. (Fig 2C, upper right panel), indicating a relief of translation inhibition.

Integrated stress response inhibitor (ISRIB) acts downstream of eIF2α in the PERK pathway by preventing p-eIF2α from binding and inhibiting eIF2B [45]. Therefore, eIF2B can recycle eIF2-GDP to active eIF2-GTP, and translation initiation can still occur, despite the levels of p-eIF2α remaining unchanged. Inhibition of the PERK pathway downstream of eIF2α is evident from the decrease in *Chop* transcription in MHV-infected cells treated with 2 μM ISRIB (S5D Fig).

STF-083010 (henceforth referred to as IREi) is a specific IRE1α endonuclease inhibitor that does not affect its kinase activity [46]. In MHV-infected cells treated with IREi at 60 μM (8 h p.

i., S5E Fig) the unspliced form of *Xbp1* was more prominent compared to the untreated MHV-infected cells, indicating a reduction in the endonuclease activity of this enzyme.

AEBSF, a serine protease inhibitor, prevents ER stress-induced cleavage of ATF6 resulting in inhibition of transcriptional induction of ATF6 target genes [47]. We investigated the induction of ATF6 target genes in MHV-infected cells treated with 100 μM AEBSF as previously described. As anticipated, *Calreticulin* and *Grp94* transcription was greatly reduced in AEBSF-treated cells (S5F Fig).

Having shown these compounds effectively inhibit the UPR in the context of infection, we moved on to assess whether this could lead to an inhibition of viral replication. Cells were infected with MHV at MOI 5 and treated with the UPRi. At 8 h p.i., tissue culture supernatant was harvested and released progeny quantified by plaque assay. We found modest but significant reductions in virus titres for all UPRi treatments in comparison to control cells, with fold reductions of between ~two-fold (IREi) and ~six-fold (ISRIB) (Fig 3A). This supports our hypothesis that modulation of the UPR can have antiviral effects.

Next, we investigated whether using the UPRi in combination would have a cumulative effect on virus release. We confirmed that combination treatment conditions led to reversal of the three branches of the UPR, assayed as described above (S6 Fig). Fig 3B displays virus titres from infected cells (8 h p.i.) at MOI 1 (blue) and MOI 5 (red), treated with different UPRi combinations. Reductions in virus titre ranged from ~four-fold, in cells incubated with PERKi and ISRIB (both targeting the PERK-eIF2α-ATF4 branch), to ~40- and ~100-fold (MOI 5 and 1 respectively), in cells treated with IREi and AEBSF (targeting the IRE1α and the ATF6 pathways). This was confirmed by western blotting, demonstrating a striking decrease in N protein levels for treatment combinations where virus titres were lowest (Fig 3C). We note that some treatment conditions largely involving ISRIB or PERKi, led to an increase in N protein levels compared to mock-treated cells. We hypothesise that this is due to alleviation of p-eIF2α-mediated translation inhibition, facilitating increased translation of viral proteins, although this does not result in increased virus release. In addition, cell monolayers infected with MHV in the presence of IREi and AEBSF showed delayed cytopathic effect, as indicated by reduced syncytium formation, likely due to lower virus production (Fig 3D).

## Mechanistic analysis of the UPR activation by SARS-CoV-2 proteins

Having established the use of UPRi as a potential antiviral strategy, we moved on to study UPR activation by SARS-CoV-2, initially assaying the cellular response to individual virus proteins in the context of transfection.

UPR activation associated with individual proteins from other CoVs in the *Betacoronavirus* genus has been investigated previously [30,48–50]. The response to the spike (S) protein has been the most thoroughly characterised, with SARS-CoV and human CoV HKU1 S activating the PERK-eIF2α-ATF4 branch [48,51] and MHV S activating the IRE1α-XBP1 pathway [30]. Additionally, SARS-CoV ORF3a, ORF6, ORF7a, ORF8ab or ORF8b proteins were able to activate the UPR [48–50,52,53], while the SARS-CoV E protein had an inhibitory effect [54]. Of these UPR-stimulatory accessory proteins, the response was characterised in detail for ORF3a, which activates the PERK-eIF2α-ATF4 branch [49], and ORF8ab, which activates the ATF6 branch [50]. To investigate potential UPR-stimulatory proteins in SARS-CoV-2, we selected the three proteins whose activation of the UPR had been best-characterised in other members of the genus: S, ORF3a and ORF8. These proteins are divergent from their SARS-CoV counterparts (76%, 72% and 26% amino acid identities, respectively), so there may be differences in UPR activation that would add to our understanding of the relationship between CoVs and the UPR. To investigate the response to these proteins, we expressed C-terminally-tagged S

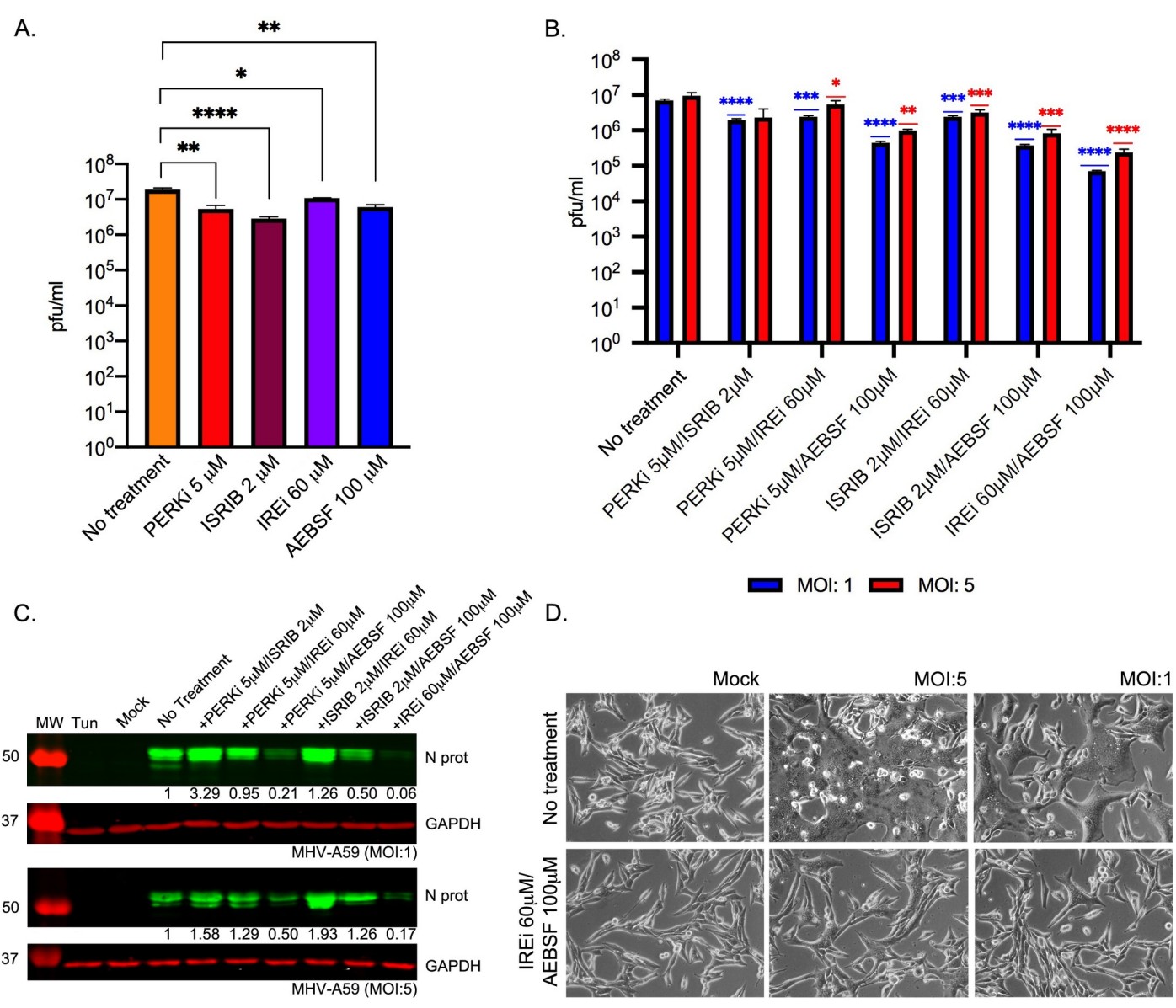

**Fig 3. Effect of UPR inhibitors on MHV replication. (A)** MHV-infected cells (MOI 5) were treated with UPR inhibitors (5 μM PERKi, 2 μM ISRIB, 60 μM IREi, or 100 μM AEBSF). The inhibitors were added to the cells immediately after the virus adsorption period and maintained in the medium until cells were harvested 8 h later. Plaque assays were performed with serial dilutions of the supernatant containing released virions from 17 Cl-1 cells infected with MHV-A59 in the presence or absence of the UPR inhibitors. Values show the mean averages of the titration of three biological replicates. Error bars represent standard errors. **(B-D)** MHV-infected cells (MOI 1 and MOI 5) were treated with dual combinations of the UPR inhibitors. The inhibitors were added to the cells immediately after the virus adsorption period and maintained in the medium until cells were harvested 8 h later. **(B)** Released virions were quantified as described in A. **(C)** Western blot analysis of MHV N protein. GAPDH was used as a loading control. Protein band quantifications for N protein, normalised by GAPDH and given relative to untreated/infected cells, are provided below. Immunoblots are representative of three biological replicates. **(D)** Representative images of mock- and MHV-infected cells at 8 h p.i. under no-drug or IREi 60 μM/AEBSF 100μM treatment conditions. All *t*-tests were two-tailed and did not assume equal variance for the two populations being compared ($^*p < 0.05$, $^{**}p < 0.01$, $^{***}p < 0.001$, $^{****}p < 0.0001$). All *p*-values are from comparisons with the respective untreated control at the same MOI.

(S-HA), ORF3a (ORF3a-FLAG) and ORF8 (ORF8-FLAG) proteins in human embryonic kidney cells (HEK-293T cells). N, a structural protein which is not documented as activating the UPR, was over-expressed as a negative control (N-FLAG).

ER stress, assessed by the induction of HERP and BiP, was induced by SARS-CoV-2 S but not N (S7A Fig). The PERK-eIF2α-ATF4 branch was activated from 24 h p.t. onwards, as indicated by the phosphorylation of eIF2α and the detection of ATF4 (S7A Fig), although phosphorylation of PERK was not clearly evident. The activation of this pathway was further confirmed by the increase in *CHOP* transcription compared to mock-transfected cells (S7B Fig). The amino terminus of ATF6 (ATF6-Nt) was detected in S-transfected cells from 24 h p.t. onwards (S7A Fig), indicating activation of the ATF6 branch. Activation of the IRE1α pathway is also evident from an increase in the spliced form of *XBP1* in S protein-transfected cells (S7A Fig). Contrary to previous findings for SARS-CoV, this indicates that the expression of the SARS-CoV-2 S protein is sufficient to induce all three major signalling pathways of the UPR.

In the case of SARS-CoV-2 ORF8 transfection, IRE1α-XBP1 and ATF6 were the main pathways induced (S7C Fig), again contrasting with findings for SARS-CoV [50]. Although a slight activation of ATF4 was observed in ORF8-transfected cells at 36 h p.t. (S7C Fig), this was not accompanied by PERK nor eIF2a phosphorylation, and induction of *CHOP* transcription was lower than in S protein-transfected cells (S7B Fig). SARS-CoV-2 ORF3a transfection did not induce any of the branches of the UPR (S7C Fig).

We then asked whether the UPR induction caused by SARS-CoV-2 S and ORF8 overexpression could be reversed by treatment with UPRi. This was assessed for each inhibitor individually (S8 Fig). Additionally, we tested this using a combination treatment condition (Fig 4), for which we selected IREi/AEBSF as this gave the most promising reduction in viral titre during MHV infection (Fig 3B). Treatment of SARS-CoV-2 S- and ORF8-transfected cells with IREi/AEBSF reduced expression of HERP and BiP to levels comparable to mock-transfected cells (Fig 4A, 36 h p.t.). This indicates the treatment successfully reversed the UPR activation by the two viral proteins. PERK pathway inhibition was evident in treated cells from the reduction in PERK and eIF2α phosphorylation (Fig 4A); however, ATF4 levels appeared to be slightly increased under these conditions, as was induction of its target gene *CHOP* (S8C Fig). ATF4 induction in the presence of IREi has been previously described [55]. Inhibition of the ATF6 and the IRE1α-XBP1 pathways was also evident, as very little ATF6-Nt and *XBP1-s* were present in IREi/AEBSF treated cells (Fig 4A and 4B).

In summary, over-expression of the S and the ORF8 proteins of SARS-CoV-2 is sufficient to activate the three branches of the UPR, and this can be reversed by UPRi treatment. It should be noted that, due to potential differences such as protein sub-cellular localisation, expression levels or available interaction partners, there may be differences in the UPR-stimulatory roles of these proteins in this system compared to infection. Future experiments using knockout mutant viruses could extend observations on the role of ORF8 to the context of infection, however S is essential for infectivity, making such experiments challenging.

## Induction of the UPR in SARS-CoV-2-infected cells

We went on to study UPR activation in the context of SARS-CoV-2 infection. Vero CCL81 cells were infected at MOI 1 and harvested at 24 and 48 h p.i., representing the exponential phase of viral replication without the cell viability being notably compromised [56]. Lysates were analysed as above. As shown in Fig 5A, the PERK-eIF2α-ATF4 branch was activated at 48 h p.i. as indicated by increased phosphorylation of PERK and eIF2α. This was further confirmed by the induction of *CHOP* in infected cells (S9A Fig). Detection of ATF6-Nt (Fig 5A) demonstrates that the ATF6 pathway is also activated during the course of infection. In addition, activation of the IRE1α pathway was evident from an increase in the spliced form of *XBP1* in SARS-CoV-2-infected cells (Fig 5A). These findings were verified in a more

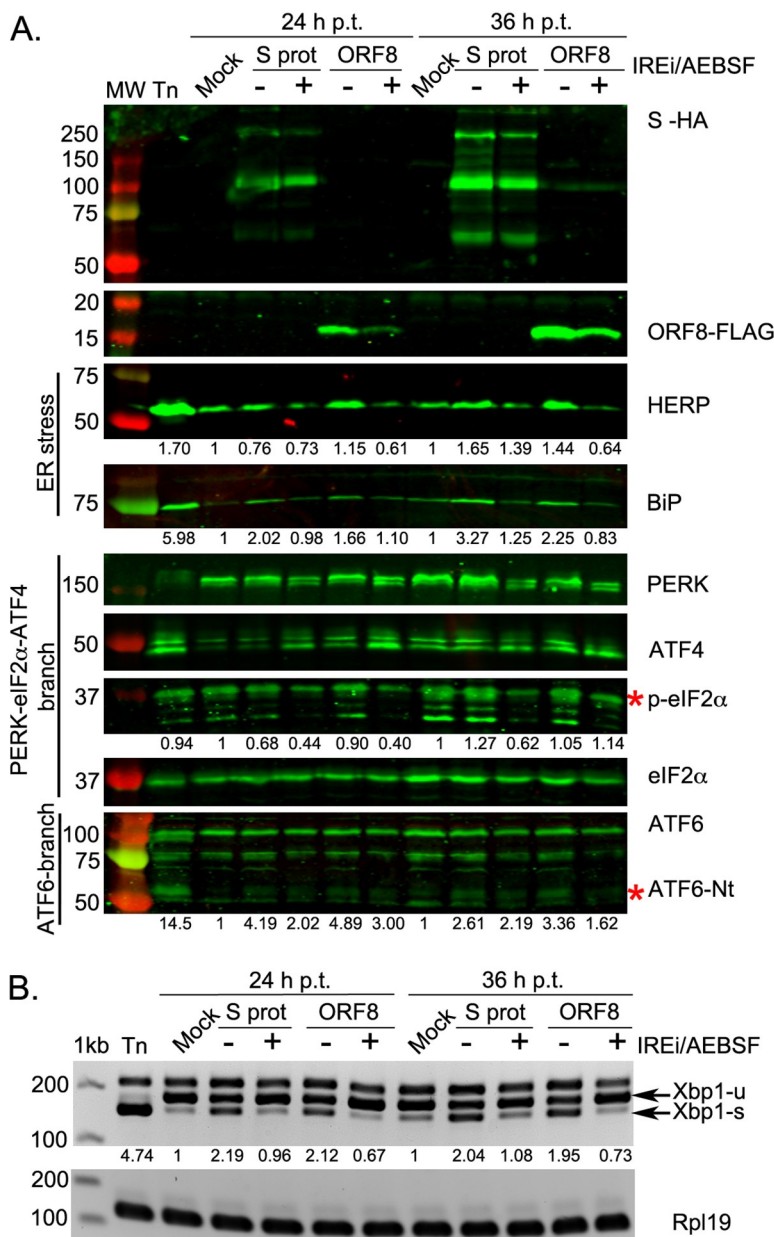

**Fig 4. Mechanistic analysis of UPR activation by SARS-CoV-2 proteins.** HEK-293T cells were transfected with plasmids encoding SARS-CoV-2 S (S-HA) or ORF8 (ORF8-FLAG), mock-transfected, or treated with tunicamycin (Tn). At 8 h p.t., cells were treated with 60 μM IREi and 100 μM AEBSF and then harvested at 24 and 36 h p.t. **(A)** Western blot analysis of ORF8-FLAG, S-HA, HERP, BiP, PERK, ATF4, p-eIF2α and ATF6 proteins. The specific p-eIF2α and ATF6-Nt bands are indicated with a red asterisk. Protein band quantifications for HERP, BiP, p-eIF2α and ATF6-Nt, normalised by eIF2α as a loading control and given relative to the mock, are provided below the respective immunoblots. **(B)** RT-PCR analysis of *XBP1-u* and *XBP1-s* mRNAs, performed as described in Fig 2D. Immunoblots and agarose gels are representative of three biological replicates. "h p.t." = hours post-transfection.

physiologically relevant context by assaying SARS-CoV-2-infected Calu3 cells (a human lung cell line) at 24 h p.i. (MOI 1), as a model for the primary site of SARS-CoV-2 infection [56,57] (Fig 5B). We conclude that SARS-CoV-2 infection induces all three branches of the UPR.

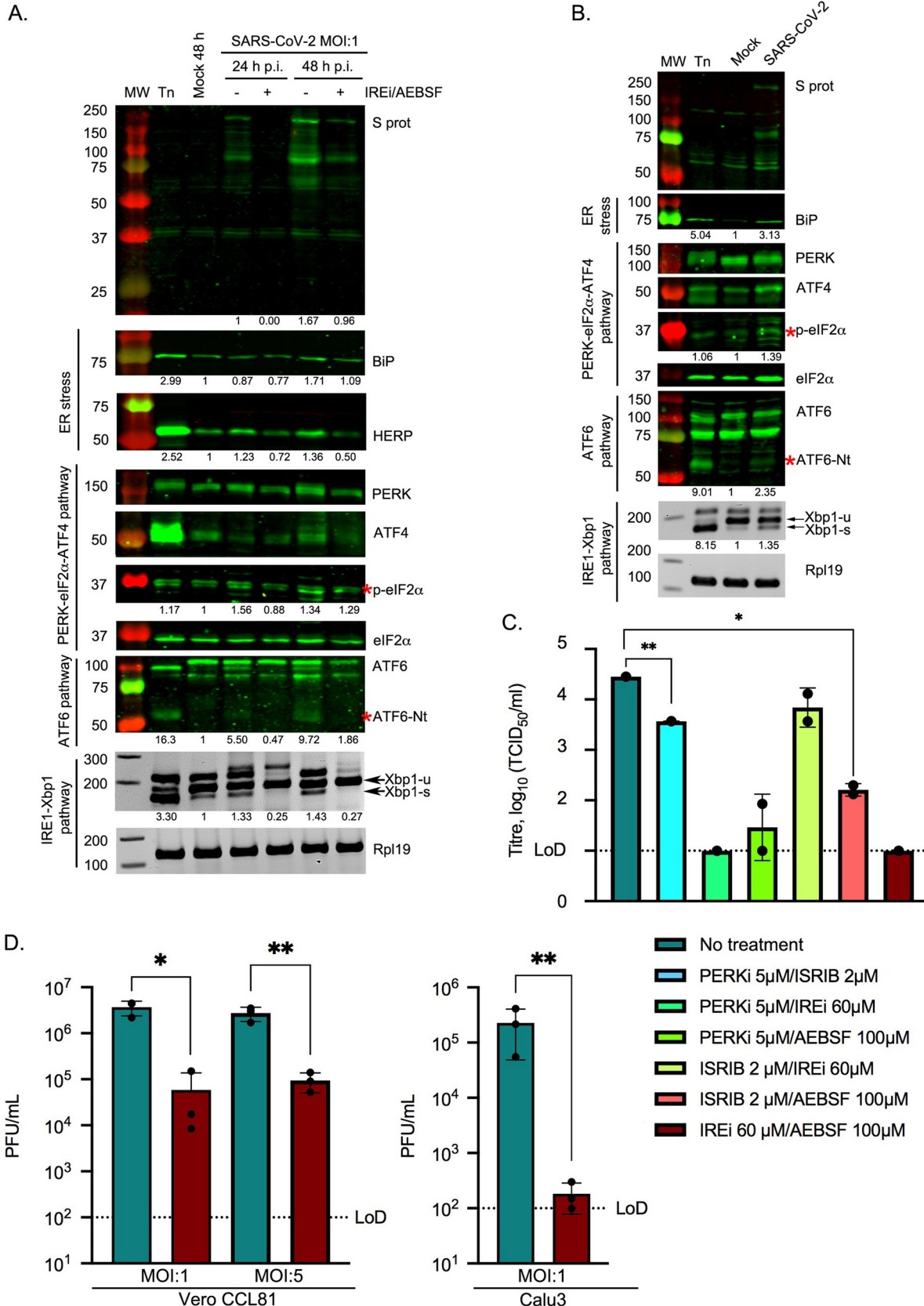

**Fig 5. Induction of the UPR in SARS-CoV-2 infected cells and the effect of UPRi.** Vero CCL81 cells **(A)** or Calu3 cells **(B)** were incubated in the presence of tunicamycin (2 μg/ml) or infected with SARS-CoV-2 (MOI 1). Infected Vero CCL81 cells were treated with 60 μM IREi and 100 μM AEBSF immediately after the virus adsorption period and inhibitors were maintained in the medium until cells were harvested 24 and 48 h later. Infected Calu3 cells were harvested at 24 h p.i. **(A-B)** Western blot analysis (upper) of SARS-CoV-2 S, BiP, HERP, PERK, ATF4, p-eIF2α and ATF6 proteins from cell lysates of Vero CCL81 cells **(A)** or Calu3 **(B)** infected cells. The specific p-eIF2α and ATF6-Nt bands are indicated by red asterisks. Protein band quantifications, normalised by eIF2α as a loading control and given relative to the mock, are provided below the respective immunoblots. RT-PCR analysis of *XBP1-u* and *XBP1-s* mRNAs (lower), performed as described in Fig 2D. Immunoblots and agarose gels are representative of three biological replicates. **(C)** $TCID_{50}$ assays were performed with serial dilutions of the supernatant containing released virions from Caco2 cells infected with SARS-CoV-2 (MOI 0.01) for 48 h in the presence or absence of the indicated UPRi combinations. **(D)** Plaque assays were performed with serial dilutions of the supernatant containing released virions from Vero CCL81 or Calu3 cells infected with SARS-CoV-2 (MOI 1 and MOI 5) for 24 h in the presence or absence of 60 μM IREi and 100 μM AEBSF. Values show the mean averages of the titration of three biological replicates. Error bars represent standard errors. All *t*-tests were two-tailed and did not assume equal variance for the two populations being compared (* $p < 0.05$, ** $p < 0.01$). Replicates with titres below the limit of detection (LoD) were excluded from *t*-tests, precluding some conditions from statistical assessment.

## Effect of the IREi/AEBSF combination on SARS-CoV-2 infection

Next, we investigated whether the previously described UPRi combinations could also be used as potential antiviral drugs against SARS-CoV-2. The gastrointestinal tract is known to be a site of SARS-CoV-2 infection *in vivo* [58] so we used Caco2 cells, human intestinal cells shown to be permissive for SARS-CoV-2 infection [56,57]. The UPRi compounds were very well-tolerated by Caco2 cells (S9B Fig). Cells were infected with SARS-CoV-2 at MOI 0.01 and treated with the different UPRi combinations. Supernatants were harvested at 48 h p.i. and released virions quantified by $TCID_{50}$ assay (Fig 5C). Reductions in virus titre were observed and these were generally much greater than those seen for MHV (MOI 1 and 5, Fig 3B), with both the PERKi/IREi and IREi/AEBSF combinations reducing virus titres to below the limit of detection.

As the IREi/AEBSF combination had the greatest inhibitory activity against both MHV and SARS-CoV-2, we tested whether this combination could inhibit SARS-CoV-2 infection at a higher MOI in Vero CCL81 and Calu3 cells. The cytotoxicity profile of these compounds in both cell lines was assayed (S9C and S9D Fig). Cells were infected at MOI 1 or MOI 5 and virus titres assessed by plaque assays at 24 h p.i. Incubation of Vero cells with IREi/AEBSF led to a statistically significant ($p = 0.0241$ for MOI 1 and $p = 0.0033$ for MOI 5) ~100-fold reduction in virus titre (Fig 5D, left). In Calu3 cells, IREi/AEBSF treatment had an even greater antiviral effect, reducing released virions by ~1,000-fold ($p = 0.0017$) to at or around the limit of detection (Fig 5D, right).

Detailed analysis of the activation of the three UPR pathways under the IREi/AEBSF treatment condition was performed in SARS-CoV-2-infected Vero CCL81 cells at 24 and 48 h p.i. (Figs 5A and S9A). Interestingly, in both SARS-CoV-2- and MHV-infected cells, the IREi/AEBSF combination was not only able to prevent activation of the IREi and ATF6 pathways, but also the PERK-eIF2α-ATF4 branch, as indicated by reduced phosphorylation of PERK and eIF2α (Figs 5A and S6) and reduced transcription of *CHOP* (S6C and S9A Figs). This may be due to the inhibition of viral replication leading to a reduced ER load, as opposed to specific inhibition of the PERK pathway. This is supported by the observation of a striking decrease in viral protein levels in infected cells treated with IREi/AEBSF (Figs 3C and 5A), consistent with reduced viral replication. This reversal of CoV-induced UPR activation by the UPRi suggests that the antiviral activity of these compounds can be attributed, at least in part, to specific inhibition of the UPR, a pathway which is evidently required for efficient viral replication.

In addition to its role in UPR inhibition, AEBSF is a relatively promiscuous arylsulfonylchloride serine protease inhibitor. It has been reported to inhibit TMPRSS2 [59,60], a host serine protease essential for SARS-CoV-2 cell entry [57]. To test whether AEBSF treatment inhibits SARS-CoV-2 cell entry, we transfected HEK-293T cells with TMPRSS2 and ACE2, the

SARS-CoV-2 cell entry receptor [61] and incubated them with lentiviral particles pseudotyped with the SARS-CoV-2 S protein (S9E Fig). No significant inhibition of viral entry was observed upon treatment with 100μM AEBSF for 4 hours, suggesting that the antiviral activity of AEBSF is predominantly due to its inhibition of the UPR.

### The effect of alternative UPR inhibitors on CoV infection

To confirm that the inhibition of virus production caused by IREi/AEBSF treatment is due to specific inhibition of the UPR, we employed two alternative compounds to target the same pathways. Ceapin-A7 selectively inhibits trafficking of ATF6 from the ER to the Golgi, thereby preventing its cleavage and activation [62–64]. KIRA8 (kinase-inhibiting RNase attenuator 8) specifically inhibits oligomerisation of IRE1α preventing activation of its RNase activity [65–67]. Ceapin-A7 at 15 μM and KIRA8 at 10 μM were very well-tolerated in 17 Cl-1, Vero CCL81 and Calu3 cells (S10 Fig), with improved metabolic and viability profiles compared to IREi/AEBSF.

In the context of MHV infection, we verified that Ceapin-A7 and KIRA8 specifically inhibit their target branches of the UPR, both individually (S11A Fig), and when used in combination (S11B Fig). Further, both compounds significantly reduced the titre of virions released from MHV-infected 17 Cl-1 cells at 8 h p.i. (S11C Fig), consistent with the observed reductions in N protein (S11A and S11B Fig) and viral RNA (vRNA) abundance (S11B Fig). These results confirm that specific inhibition of the ATF6 or IRE1α branch of the UPR is sufficient to inhibit MHV replication, with combination treatment producing the greatest effect.

Next, we tested Ceapin-A7 and KIRA8 in the context of SARS-CoV-2 infection. Vero cells were infected with SARS-CoV-2 at MOI 5, treated with the inhibitors individually or in combination, and harvested at 24 and 48 h p.i. (Fig 6). Successful inhibition of the UPR was confirmed (Fig 6A) and the effect on virion release was assayed (Fig 6B). Both Ceapin-A7 and KIRA8 significantly reduced the titre of released virions. KIRA8 caused a greater reduction than Ceapin-A7, while the combination of both inhibitors was the most effective, reducing released virions by ~60-fold and ~500-fold at 24 and 48 h p.i., respectively. Striking reductions in abundance of viral RNA (Fig 6C) and proteins (Fig 6A) upon Ceapin-A7/KIRA8 treatment provides further evidence of the strong inhibitory effect of these UPRi on SARS-CoV-2 replication. Ceapin-A7/KIRA8 treatment was ~4-fold more effective at reducing released virions than IREi/AEBSF at 24 h p.i. ($p$ = 0.0039). At 48 h p.i., this increased to a ~135-fold difference in effectiveness ($p < 0.0001$). Further experiments in Calu3 cells (MOI 5) reveal that Ceapin-A7/KIRA8 treatment significantly reduced virion release at 24 h p.i., to the same extent as IREi/AEBSF treatment (S11D Fig). The similar, or even greater, reductions in virion release caused by Ceapin-A7/KIRA8 compared to the more promiscuous inhibitor combination, IREi/AEBSF, demonstrate that the antiviral activity of these compounds is due to specific inhibition of the UPR.

Taken together, these results show that highly specific inhibition of the ATF6 and IRE1α branches of the UPR significantly inhibits CoV replication, making the UPR a promising antiviral target.

### Discussion

This study reveals that all three branches of the UPR are activated upon MHV and SARS-CoV-2 infection, and highlights this as a very prominent pathway in the host response. The UPR was the most significantly enriched Reactome pathway associated with genes transcriptionally up-regulated during MHV infection and, consistent with previous studies, we show activation of all three branches of the UPR by MHV [28,30]. Confirming the importance of

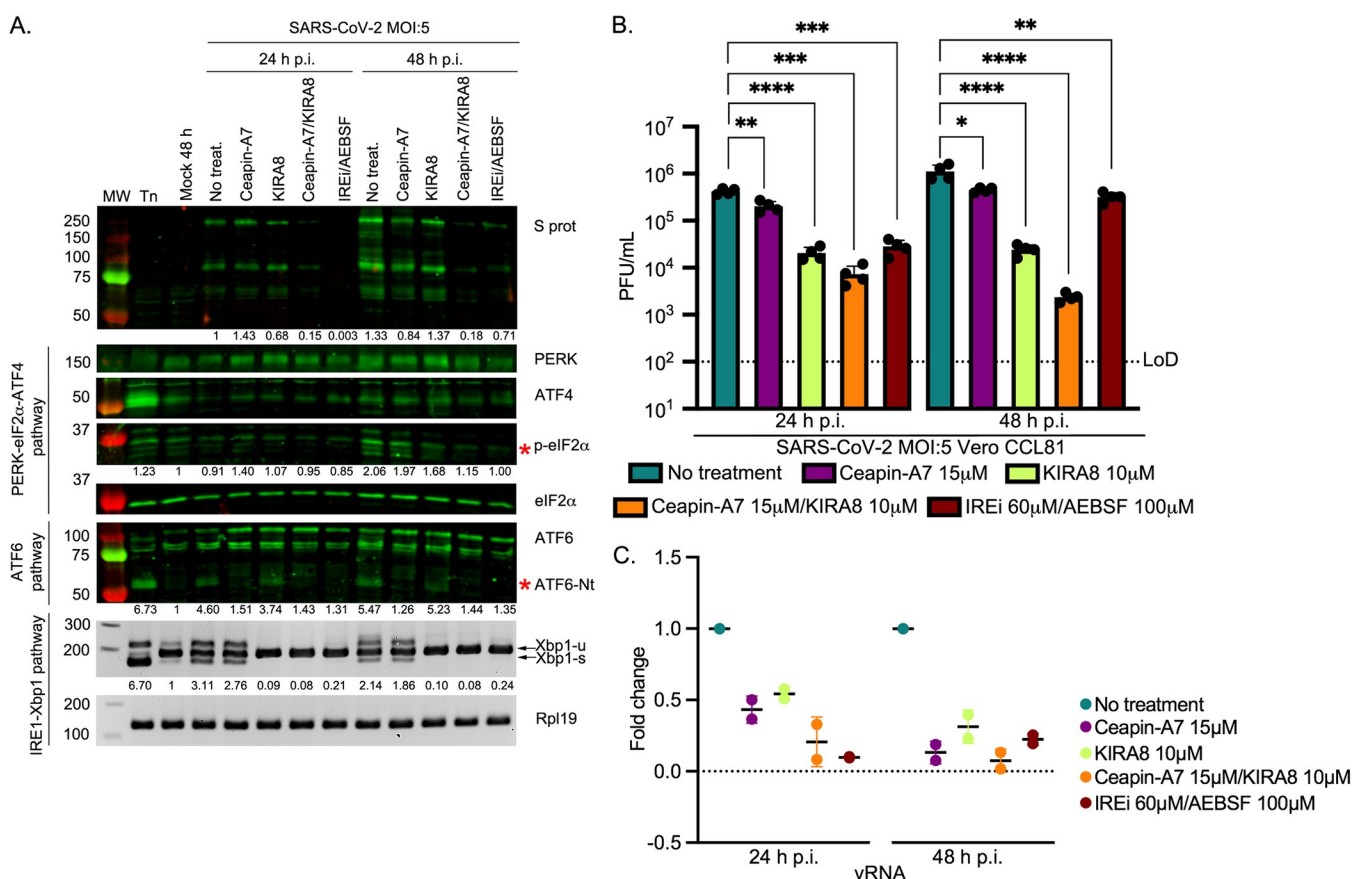

**Fig 6. Effect of specific inhibition of IRE1α and ATF6 pathways on SARS-CoV-2 replication.** Vero CCL81 cells were incubated in the presence of tunicamycin (2 μg/ml) or infected with SARS-CoV-2 (MOI 5) and treated with 15 μM Ceapin-A7 and 10 μM KIRA8 as individual treatments or in combination. The inhibitors were added to the cells immediately after the virus adsorption period and maintained in the medium for 24 or 48 h. **(A)** Western blot analysis (upper) of SARS-CoV-2 S, PERK, ATF4, p-eIF2α and ATF6. The specific p-eIF2α and ATF6-Nt bands are indicated by red asterisks. Protein band quantifications, normalised by eIF2α as a loading control and given relative to the mock, are provided below the respective immunoblots. RT-PCR analysis of *XBP1-u* and *XBP1-s* mRNAs (lower), performed as described in Fig 2D. Immunoblots and agarose gels are representative of three biological replicates. **(B)** Plaque assays were performed with serial dilutions of the supernatant containing released virions at 24 and 48 h p.i. Values show the mean averages of the titration of four biological replicates. Error bars represent standard errors. All *p*-values are from comparisons with the respective untreated control, with *$p < 0.05$, **$p < 0.01$, ***$p < 0.001$ and ****$p < 0.0001$. **(C)** RT-qPCR of vRNA from two biological replicates of Vero CCL81 cells infected and treated as described above. Data are normalised as described in Fig 2B.

this in SARS-CoV-2 infection, ER-related GO/KEGG terms are enriched in the differentially expressed genes lists of several proteomics/transcriptomics studies on SARS-CoV-2-infected cells [32,68–71]. This is also a very prominent theme in proteomics studies identifying host interaction partners of SARS-CoV-2 proteins, in which ER proteins are reproducibly found [70,72,73]. In one such study, "response to endoplasmic reticulum stress" was the most highly enriched biological process GO annotation associated with the host interaction partners [73]. This suggests that SARS-CoV-2, like other CoVs [74–76], enacts a finely tuned modulation of the UPR that may involve direct interactions with its components. Despite this, the activation of the three branches of the UPR by SARS-CoV-2 has not been previously described, although it has been characterised for other CoVs [6,32,51,74,77–80] including the closely related SARS-CoV [30,48–50,52–54,81,82]. Here we show that, like MHV, SARS-CoV-2 infection induces all three branches of the UPR, in contrast to results from SARS-CoV infection, in which only the PERK branch was activated [30,54,81].

Over-expression of the individual SARS-CoV-2 S or ORF8 proteins initiates UPR signalling. S protein was found to induce all three branches of the UPR in contrast to the counterpart protein of SARS-CoV, which appears to induce exclusively the PERK pathway [48]. Similarly, we identify ORF8 of SARS-CoV-2 as an inducer of both the IRE1α and ATF6 branches of the UPR, whereas the SARS-CoV equivalent has been shown to activate only ATF6 [50]. These differences can partly be explained by sequence divergence between the two viruses [83]. SARS-CoV-2 ORF8 maintains only 26% amino acid identity with SARS-CoV ORF8ab [84], so the two proteins may have very different relationships with the UPR. For example, SARS-CoV-2 ORF8 lacks the VLVVL motif that causes SARS-CoV ORF8 (specifically ORF8b) to aggregate and trigger intracellular stress pathways [53]. SARS-CoV ORF8ab was shown to mediate activation of the ATF6 pathway through a direct interaction with the ATF6 ER-lumenal domain [50], although it is undetermined whether the corresponding interaction occurs with SARS-CoV-2 ORF8. Recent proteomics-based interactome studies have identified interactions between SARS-CoV-2 ORF8 and several ER quality control proteins [70,72], which could contribute to the ORF8-induced UPR induction observed in our study. Alterations to this key UPR modulator have important ramifications: mutation or deletion of ORF8 in naturally occurring strains of SARS-CoV and SARS-CoV-2 correlate with milder disease and, in the latter case, lower incidence of hypoxia [85–88].

Here, we also demonstrate the importance of UPR activation to CoV infection by showing that pharmacological inhibition of the UPR leads to significant reductions in titres of virions released from MHV- and SARS-CoV-2-infected cells. Simultaneous inhibition of the IRE1α and ATF6 pathways was particularly effective, reducing virus titres by up to ~500–1,000-fold. Some of the compounds we used to target these pathways (i.e. STF-083010 and KIRA8 against IRE1α and AEBSF against ATF6) have been extensively used in preclinical studies for neurodegenerative diseases, autoimmune diabetes, cancer and pulmonary fibrosis [46,65,89–93]. Thus, simultaneous inhibition of IRE1α and ATF6 pathways using these drugs represents a promising antiviral strategy that could rapidly progress into a clinical trial.

To date, the development of antivirals against SARS-CoV-2 has focused on drugs targeting virus replication, such as remdesivir. However, these antiviral therapies do not take into account that the pathophysiology associated with COVID-19 is mostly related to an aberrant cellular response. In some clinical manifestations of COVID-19, an exacerbated UPR could play a key role [94–96]. For example, activation of ER stress and the UPR is one of the major triggers of endothelial dysfunction [97,98], which is associated with acute respiratory distress syndrome (ARDS) [99], a diffuse inflammatory lung injury present in 20–67% of hospitalised patients [100,101]. Other clinical manifestations of COVID-19 such as thromboembolism, cerebro- and cardiovascular diseases and neurological complications, are also associated with endothelial dysfunction [102]. Furthermore, a recognised sequela of COVID-19 is pulmonary fibrosis [103], which can develop in up to 17% of COVID-19 patients [104]. Pulmonary fibrosis is a severe form of interstitial lung disease characterised by progressive dyspnea, hypoxemia, and respiratory failure due to the presence of patchy areas of fibrotic tissue. ER stress and UPR activation are known to be involved in the development and progression of this fibrotic disease [105]. This suggests that UPR activation in response to SARS-CoV-2 infection contributes to the lung pathophysiology associated with COVID-19. Therefore, the UPR inhibitors used in this study could have a dual therapeutic effect, not only contributing to the reduction of viral burden in patients, but also diminishing the pathophysiology associated with COVID-19. In addition, the idea of targeting an exaggerated cellular response instead of the virus itself substantially reduces the chances of generating virus escape mutants.

## Materials and methods

### Cells and viruses

Murine 17 clone 1 (17 Cl-1), Calu3 (ATCC, HTB-55, a kind gift from Prof Frank Kirchhoff, Institute of Molecular Virology, Ulm University Medical Center) and Vero (ATCC, CCL81) cells were maintained in Dulbecco's modification of Eagle's medium supplemented with 10% (vol/vol) fetal calf serum (FCS). HEK-293T cells (ATCC, CRL-11268) were cultured in DMEM supplemented with 5% FCS. Caco2 cells were a kind gift from Dr Valeria Lulla and were maintained in DMEM supplemented with 20% FCS. All cell lines were cultured in medium containing 100 U/ml penicillin, 100 μg/ml streptomycin and 1 mM L-glutamine. Cells were incubated at 37˚C in the presence of 5% $CO_2$.

Recombinant MHV strain A59 (MHV-A59) was derived as described previously [106]. Upon reaching 70–80% confluence, 17 Cl-1 cells were infected with MHV-A59 at MOI 5 as described [9]. Vero CCL81 and Calu3 cells were infected with SARS-CoV-2 (SARS-CoV-2/human/Switzerland/ZH-UZH-IMV5/2020) at two MOIs (1 and 5) for 24 or 48 has previously described [107,108]. Caco2 cells were infected with SARS-CoV-2 (isolate hCoV-19/Edinburgh/2/2020, a kind gift from Dr Christine Tait-Burkhard and Dr Juergen Haas) at MOI 0.01 and incubated for 48 h in MEM containing 1% L-glutamine, 1% non-essential aminoacids, 1% penicillin/streptomycin and supplemented with 2% FBS.

### Ribosomal profiling and RNASeq data

17 Cl-1 cells were grown on 100-mm dishes to 90% confluency and infected with MHV-A59 at MOI 10. At the indicated time-points, cells were rinsed with 5 ml of ice-cold PBS, flash frozen in a dry ice/ethanol bath and lysed with 400 μl of lysis buffer [20 mM Tris-HCl pH 7.5, 150 mM NaCl, 5 mM $MgCl_2$, 1 mM DTT, 1% Triton X-100, 100 μg/ml cycloheximide and 25 U/ml TURBO DNase (Life Technologies)]. The cells were scraped extensively to ensure lysis, collected and triturated ten times with a 26-G needle. Cell lysates were clarified by centrifugation at 13,000 $g$ for 20 min at 4˚C. Lysates were subjected to RiboSeq and RNASeq based on previously reported protocols [9,109]. Ribosomal RNA was removed using Ribo-Zero Gold rRNA removal kit (Illumina) and library amplicons were constructed using a small RNA cloning strategy adapted to Illumina smallRNA v2 to allow multiplexing. Amplicon libraries were deep sequenced using an Illumina NextSeq500 platform. Due to the very large amounts of vRNA produced during infection, mock samples were processed separately from infected samples to avoid contamination. RiboSeq and RNASeq sequencing data have been deposited in the ArrayExpress database under the accession numbers E-MTAB-8650 (https://www.ebi.ac.uk/arrayexpress/experiments/E-MTAB-8650/) and E-MTAB-8651 (https://www.ebi.ac.uk/arrayexpress/experiments/E-MTAB-8651/).

### Computational analysis of RiboSeq and RNASeq data

Reads were trimmed for adaptor sequences, filtered for length ≥ 25 nt, and reads mapping to *Mus musculus* rRNA (downloaded from the SILVA database [110] or MHV-A59 viral RNA (GenBank accession AY700211.1) (with up to 2 mismatches) removed, as previously described [9]. The remaining reads were aligned directly to the mouse genome (FASTA and GTF gencode release M20, GRCm38, primary assembly) (with up to 2 mismatches) using STAR (parameters:--outFilterIntronMotifs RemoveNoncanonicalUnannotated--outMultimapperOrder Random) [111]. Reads on protein-coding genes were tabulated using htseq-count (version 0.9.1), covering the whole gene for differential transcription analysis (parameters: -a 0 -m union -s yes -t gene) and just the CDS for the translation efficiency analysis (parameters: -a 0

-m intersection-strict -s yes -t CDS) [112], using the GTF file from the above Gencode release as the gene feature annotation.

Differential transcription analysis was performed using DESeq2 (version 1.18.1) [10] and translation efficiency analysis with Xtail (version 1.1.5) [12]. For each analysis, low count genes (with fewer than ten counts from all samples combined) were discarded, following which read counts were normalised by the total number of reads mapping to host mRNA for that library, using standard DESeq2 normalisation. This minimises the effect of the large amount of vRNA present in infected samples. Shrinkage of the transcriptional fold changes to reduce noise in lowly-expressed genes was applied using lfcShrink (parameter: type = 'normal').

A given gene was considered to be differentially expressed if the FDR-corrected $p$ value was less than 0.05 and the fold change between the means of infected and mock replicates was greater than two. Volcano plots and transcription versus TE comparison plots were generated using R and FDR-corrected $p$ values and $\log_2$(fold change) values from the DESeq2 and Xtail analyses. All reported $p$ values are corrected for multiple testing by the Benjamini-Hochberg method. Fold changes plotted in the transcription vs TE comparison are not filtered for significant $p$ values before plotting.

To plot RNASeq and RPF profiles for specific transcripts, reads were mapped to the specified transcript from the NCBI genome assembly using bowtie [113] allowing two mismatches (parameters: -v 2,--best). Coordinates for known uORFs were taken from the literature [13,14,25] and the positions of start and stop codons in all frames determined. Read density (normalised by total reads mapping to host mRNA for each library, to give reads per million mapped reads) was calculated at each nucleotide on the transcript and plotted, coloured according to phase. Read positions were offset by +12 nt so that plotted data represent the inferred position of the ribosomal P site. Bar widths were increased to 12 nt (Fig 1E) or 4 nt (S2 Fig) to aid visibility and were plotted sequentially starting from the 5′ end of the transcript.

## Gene ontology and Reactome pathway enrichment analyses

Lists of gene IDs of significantly differentially expressed genes (S2 Table) were used for GO term enrichment analysis by the PANTHER web server under the default conditions (release 20190606, GO database released 2019-02-02) [114], against a background list of all the genes that passed the threshold for inclusion in that expression analysis. For Reactome pathway enrichment (version 69) [11], the same differentially expressed gene lists were converted to their human orthologues and analysed, both using the reactome.org web server, to determine which pathways are significantly over-represented (FDR-corrected $p$ value ≤0.05).

## Enrichment analysis for eIF2α-phosphorylation-resistant genes

Resistance to translational repression by p-eIF2α is not an existing GO term, so a list of genes reported to be p-eIF2α-resistant was constructed based on Andreev et al., 2015 [18] and references within (excluding those from IRESite, which were not found to be p-eIF2α-resistant in their study). Mouse homologues of these genes were identified using the NCBI homologene database (S4 Table). Enrichment of genes categorised as p-eIF2α-resistant amongst the genes with significantly increased translational efficiency, compared to a background of all *Mus musculus* genes included in the TE analysis with any GO annotation, was calculated using a Fisher Exact test.

## Chemicals

GSK-2606414 was a kind gift from Dr Edward Emmott and Prof Ian Goodfellow. AEBSF, STF-083010, ISRIB, Ceapin-A7 and tunicamycin were purchased from Sigma-Aldrich. KIRA8

(AMG-18) was obtained from MedChemExpress. GSK-2606414, STF-083010, ISRIB, Ceapin-A7, KIRA8 and tunicamycin were dissolved in DMSO, whereas AEBSF was dissolved in water, to the required concentrations. In all experiments, the final concentration of DMSO did not exceed 0.4% and no differences in viability were observed between untreated cells and DMSO-treated samples. Cytotoxicity after treatment with single and combined UPR inhibitors was measured using the CellTiter-Blue kit (Promega) following manufacturer's instructions and trypan blue (Sigma) exclusion assay.

## Antibodies

The following primary antibodies were used: mouse monoclonal antibodies against MHV N and S proteins (kind gifts of Dr Helmut Wege, University of Würzburg), rabbit polyclonal anti-SARS-CoV-2 spike glycoprotein antibody (ab272504, Abcam) mouse anti-GAPDH (IgM specific, G8795, Sigma-Aldrich), mouse anti-Flag (F3165, Sigma-Aldrich), rabbit anti-HA (3724, Cell Signaling Technology), rabbit anti-PERK (ab229912, Abcam), rabbit anti-HER-PUD1 (ab150424, Abcam), rabbit anti-GRP78 (BIP, ab108613, Abcam), rabbit anti-eIF2$\alpha$ (9722, Cell Signaling Technology), rabbit anti-phospho-eIF2$\alpha$ (Ser51, 9721, Cell Signaling Technology), rabbit anti-ATF4 (10835-1-AP, Proteintech), rabbit anti-ATF6 (ab203119 and ab37149, Abcam), mouse anti-S6 (2317, Cell Signaling Technology) and rabbit RPL10a (ab174318, Abcam). Secondary antibodies used for western blotting were purchased from Licor: IRDye 800CW Donkey Anti-Mouse IgG (H+L), IRDye 800CW Donkey Anti-Rabbit IgG (H+L), IRDye 680RD Goat Anti-Mouse IgG (H+L) and IRDye 680RD Goat Anti-Mouse IgM (μ chain specific).

## Plasmids and transfections

HEK-293T cells were transiently transfected with pcDNA3.1-SARS-CoV-2-S-HA (kind gift of Dr Jerome Cattin and Prof Sean Munro, MRC-LMB, Cambridge, UK), pcDNA6-SARS-CoV-2-N-FLAG, pcDNA6-SARS-CoV-2-ORF3a-FLAG and pcDNA6-SARS-CoV-2-ORF8-FLAG plasmids (kind gifts of Prof Peihui Wang, Shandong University, China) using a commercial liposome method (TransIT-LT1, Mirus). Transfection mixtures containing plasmid DNA, serum-free medium (Opti-MEM; Gibco-BRL) and liposomes were set up as recommended by the manufacturer and added dropwise to the tissue culture growth medium. Cells were harvested at 24 and 36 h post-transfection.

## Immunoblotting

Cells were lysed in 1X Laemmli's sample buffer. After denaturation at 98°C for 5 minutes, proteins were separated by 12% SDS-PAGE and transferred to nitrocellulose membranes. These were blocked (5% non-fat milk powder or bovine serum albumin in PBST [137 mM NaCl, 2.7 mM KCl, 10 mM $Na_2HPO_4$, 1.5 mM $KH_2PO_4$, pH 6.7, and 0.1% Tween 20]) for 30 min at room temparature and probed with specific primary antibodies at 4°C overnight. Membranes were incubated in the dark with IRDye-conjugated secondary antibodies diluted to the recommended concentrations in PBST for 1 h at room temperature. Blots were scanned using an Odyssey Infrared Imaging System (Licor). Quantification of protein bands (densitometry) was performed using the 'area under the curve' method in ImageJ 1.X software [115]. Relative protein expression was calculated by sequentially normalising against the loading controls (either GAPDH or eIF2$\alpha$) and then against the timepoint-matched mock for cellular proteins or the timepoint-matched infected but untreated cells for viral proteins (summary of the quantifications in S6 Table).

## Analysis of *Xbp1* splicing by RT-PCR

Total RNA was isolated from infected or transfected cells as described previously [9], and cDNA synthesised from 500 ng total RNA using M-MLV Reverse Transcriptase (Promega). Mouse or human *Xbp1* and *Rpl19* were amplified using specific primers (S5 Table). Following PCR reactions, the resulting amplicons were subjected to electrophoresis in 3% agarose gels. Quantification of *Xbp1-s* and *Xbp1-u* PCR bands was performed using the 'area under the curve' method in ImageJ 1.X software [115]. The extent of *Xbp1* splicing was quantified as *Xbp1-s* / (*Xbp1-s* + *Xbp1-u*), which represents the ratio of *Xbp1-s* to the sum of *Xbp1-s* and *Xbp1-u*, normalised against the same ratio for the timepoint-matched mock (summary of the quantifications in S6 Table).

## Quantitative real-time PCR assays

Relative levels of mouse or human *Bip*, *Chop*, *Gadd34*, *Calreticulin*, *Grp94*, MHV N and SARS-CoV-2 N in cDNA samples were determined by quantitative real-time PCR (qPCR) using a Rotor-Gene 3000 (Corbett Research). Reactions were performed in a final volume of 20 µl containing Hot Start Taq (1 U, QIAGEN), 3.5 mM $MgCl_2$, 2.5 mM deoxynucleotides, 450 nM SYBR Green dye, 500 nM relevant forward and reverse primers (S5 Table) and 1 µl of cDNA. vRNA in MHV and SARS-CoV-2 infected cells is quantified with oligonucleotides whose primer binding site is within the *N* gene (S5 Table). This will detect all canonical positive sense vRNA (i.e. genomic and all subgenomic RNAs). No-template controls were included for each primer pair, and each qPCR reaction was carried out in duplicate. Fold changes in gene expression relative to the mock were calculated by the delta delta-cycle threshold (ΔΔCt) method, and *Rpl19* was used as a normalising housekeeping gene.

## Polysome profiling

17 Cl-1 cells were infected as described above. 10 min prior to harvesting, cells were treated with cycloheximide (100 µg/ml), washed with PBS and lysed in a buffer containing 20 mM Tris HCl pH 7.5, 100 mM KCl, 5 mM MgOAc, 0.375 mM CHX, 1 mM DTT, 0.1 mM PMSF, 2 U/µl DNase I, 0.5% NP-40, supplemented with protease and phosphatase inhibitors (Thermo-Fisher Scientific). Following trituration with a 26-G needle (ten passes), lysates were cleared (13,000 *g* at 4˚C for 20 min) and the supernatants layered onto 12 ml sucrose density gradients (10–50% sucrose in TMK buffer: 20 mM Tris-HCl pH 7.5, 100 mM KCl, 5 mM $MgCl_2$) prepared in Beckman SW41 polypropylene tubes using a Gradient Master (Biocomp). Following centrifugation (200,000 *g* for 90 min at 4˚C), fractions were prepared using an ISCO fractionator monitoring absorbance at 254 nm. Proteins were concentrated from fractions using methanol-chloroform extraction and subjected to immunoblotting analysis. Polysome profiling in higher salt conditions was carried out as described above except that the lysis buffer and sucrose density gradient contained 400 mM KCl.

## Virus plaque assays

To determine MHV-A59 titres by plaque assay, 17 Cl-1 cells in 6-well plates were infected with 400 µl of 10-fold serial dilutions of sample in infection medium (Hank's balanced salt solution containing 50 µg/ml DEAE-dextran and 0.2% bovine serum albumin—BSA). After 45 min at 37˚C with regular rocking, the inoculum was removed and replaced with a 1:1 mixture of 2.4% Avicel and MEM 2X medium (20% MEM 10X, 2% non-essential aminoacids, 200 U/ml penicillin, 200 µg/ml streptomycin, 2 mM L-glutamine, 40 mM HEPES pH 6.8, 10% tryptose phosphate broth, 10% FCS and 0.01% sodium bicarbonate). Plates were incubated at 37˚C for 48 h

prior to fixing with 3.7% formaldehyde in PBS. Cell monolayers were stained with 0.1% toluidine blue to visualise plaques. SARS-CoV-2 plaque assays were performed as previously described [107]. Experiments were conducted using three biological repeats.

## TCID$_{50}$ assays

SARS-CoV-2 replication was assessed using a 50% tissue culture infective dose (TCID$_{50}$) assay in Vero E6 cells. Supernatant derived from infected Caco2 cells was subjected to 10-fold serial dilutions. At 72 h p.i., cells were fixed and stained as previously indicated. Wells showing any sign of cytopathic effect (CPE) were scored as positive.

## Statistical analysis of virus titre results

Data were analysed in GraphPad Prism 9.0 (GraphPad software, San Diego, CA, USA). Values represent mean ± standard deviation. Statistical significance was evaluated using two-tailed $t$-tests on $\log_{10}$(virus titre) data, which did not assume equal variances for the two populations being compared, to calculate the $p$-values. Differences as compared to the control with $p$ value $\leq 0.05$ were considered as statistically significant, with $^{*}p < 0.05$, $^{**}p < 0.01$, $^{***}p < 0.001$ and $^{****}p < 0.0001$.

## Supporting information

**S1 Fig. Quality control indicates sequencing data are of high quality. (A)** Length distribution of positive-sense reads mapping within host CDSs. For RiboSeq libraries (pink) the characteristic sharp peak at 28–29 nt is observed, reflective of the length of mRNA protected from RNase I digestion. Read lengths of RNASeq libraries (green) are determined by alkaline hydrolysis and gel purification size selection (25–34 nt), leading to a broader length distribution. **(B)** Percentage of reads (all read lengths) attributed to each phase, for positive-sense reads mapping within host CDSs. Phases correspond to which position within the codon the 5′ end of the read maps to (0: purple, 1: blue, 2: yellow). The 5′ end coordinate of RiboSeq reads is influenced by the position of the translating ribosome, leading to a clear dominance of the 0 phase. For RNASeq reads the 5′ end coordinate is determined by alkaline hydrolysis so does not result in a dominant phase. **(C)** Distribution of host mRNA-mapping reads relative to start and stop codons. Only transcripts with an annotated CDS of at least 150 codons, 5′ UTR of at least 60 nt, and 3′ UTR of at least 60 nt were included in the analysis. The total number of positive-sense reads from all these transcripts mapping to each position was plotted, with an offset of +12 relative to the 5′ end coordinate to represent the inferred ribosomal P site. Although ribosomal P site position is not relevant to RNASeq reads, these were also plotted with a +12 nt offset to facilitate comparison. Data are coloured according to phase as in B. For RiboSeq libraries there is clear triplet periodicity visible across the CDS, reflective of the length of a codon, and a large peak corresponding to terminating ribosomes—characteristic of samples harvested without drug pre-treatment. Very few RiboSeq reads map to the UTRs (and particularly the 3′ UTR), indicating very little contamination of the mRNA fraction with non-ribosome-protected-fragment reads. As expected, for RNASeq libraries the coverage does not differ greatly between the CDS and UTRs.
(TIF)

**S2 Fig. Distribution of reads mapping to specific host genes of interest.** Analysis of RPFs (mock and MHV-infected samples plus tunicamycin-treated sample) and RNASeq reads (mock and MHV-infected samples) mapping to *Xbp1-u* (NCBI RefSeq mRNA NM_013842). Cells were infected with MHV-A59 or mock-infected and harvested at 5 h p.i. or 8 h p.i.

(libraries from Fig 1D and 1E). One sample was treated with 2 μg/ml tunicamycin, a pharmacological inducer of all three branches of the UPR, as a positive control. Reads are plotted at the inferred position of the ribosomal P site and coloured according to phase: pink for 0, blue for +1, yellow for +2. The 5′ end position of RNASeq reads is not determined by ribosome position and therefore should not show a dominant phase. The main ORF (0 frame) is shown at the top in pink, with start and stop codons in all three frames marked by green and red bars (respectively) in the three panels below. The yellow rectangle in the +2 frame indicates the extended ORF that results from splicing by IRE1. Reads resulting mainly from translation of the spliced *Xbp1-s* isoform can be seen in yellow (+2 phase), downstream of the main ORF annotated stop codon. Dotted lines serve as markers for the start and end of the features in their matching colour. Read densities are plotted as reads per million host-mRNA-mapping reads. Bar widths were increased to 4 nt to aid visibility, and therefore overlap, and were plotted sequentially starting from the 5′ end of the transcript.
(TIF)

**S3 Fig. ATF6 pathway activation in MHV-infected cells.** 17 Cl-1 cells were incubated in the presence of tunicamycin (2 μg/ml) or infected with MHV-A59 (MOI 5) and harvested at 2.5, 5 and 8 h. **(A)** Cell lysates were separated by 12% SDS-PAGE and immunoblotted using anti-ATF6 (1:1000, Abcam ab203119, upper), anti-ATF6 (1:1000, Abcam ab37149, middle). GAPDH was used as loading control. Representative images of fixed and permeabilised cells treated with tunicamycin for 6 h **(B)** or infected with MHV for 8 h **(C)** and incubated with anti-ATF6 (1:500, Abcam ab37149, red) and anti-S protein (green). Nuclei are counterstained with DAPI (blue). Images were taken in an Evos FLII microscope at 60X magnification. Scale bar: 100 μm. **(D)** Analysis of RPFs and RNASeq reads mapping to *Bip* (NM_022310). Plot constructed as described in S2 Fig. Note that in order to properly visualise RPFs across the ORF, the y-axis has been truncated at 400 reads per million host-mRNA-mapping reads for the RiboSeq samples, leaving some RPF counts for tunicamycin-treated cells and MHV-infected cells off-scale.
(TIF)

**S4 Fig. Cytotoxicity assays of UPR inhibitors in 17 Cl-1 cells.** 17 Cl-1 cells were treated with the different UPR inhibitors at the indicated concentrations. Experiments were performed in triplicate using CellTiter-Blue Cell Viability Assay to assess metabolic capacity (dashed line represents 70% threshold) **(A)** and in duplicate using trypan blue exclusion assay to assess cell proliferation and viability **(B)** in treatment conditions involving IREi or AEBSF. Cell viability in all cases tested was greater than 85% (dotted line). Percentages are given relative to untreated cells. Error bars represent standard errors.
(TIF)

**S5 Fig. Effect of UPR inhibitors on activation of the UPR during MHV infection.** MHV-infected cells (MOI 5) were treated with UPR inhibitors (5 μM PERKi, 2 μM ISRIB, 60 μM IREi, or 100 μM AEBSF). The inhibitors were added to the cells immediately after the virus adsorption period and maintained in the medium until cells were harvested 8 h later. **(A)** Western blot analysis of MHV N protein, PERK, ATF4, BiP, p-eIF2α and eIF2α as loading control. Protein band quantifications (performed as described for Fig 2) are provided for MHV N protein and p-eIF2α and immunoblots are representative of three biological replicates. **(B)** 17 Cl-1 cells infected with MHV-A59 and treated with 0, 2.5 or 5 μM of PERKi were metabolically pulse-labelled with [$^{35}$S]Met for 1 h at 5 h p.i. Cells were lysed just after pulse and subjected to 10% SDS-PAGE followed by autoradiography. **(C)** Polysome profiling as described in Fig 2C of MHV-infected cells at 5 h p.i. treated with 5 μM of PERKi. **(D)** RT-qPCR (performed as described in Fig 2B) of *Bip* and *Chop* mRNA from two biological

replicates of MHV-infected cells treated with UPR inhibitors as described in Fig 2B. (**E**) RT-PCR analysis of *Xbp1-u* and *Xbp1-s* mRNAs, performed as described in Fig 2D. (**F**) RT-qPCR (performed as described in Fig 2B) of *Calreticulin* and *Grp94* mRNA from two biological replicates of MHV-infected cells treated with UPR inhibitors as described in Fig 2B. (TIF)

**S6 Fig. Effect of dual combinations of UPR inhibitors on activation of the three branches of the UPR during MHV infection.** MHV-infected cells were treated with the combinations of the UPR inhibitors shown in Fig 3B. The inhibitors were added to the cells immediately after the virus adsorption period and maintained in the medium until cells were harvested 8 h later. Western blot analysis (upper) of MHV N, PERK, and p-eIF2α proteins from cell lysates of MHV-infected cells at MOI 1 (**A**) and at MOI 5 (**B**). The N protein panels have been duplicated from Fig 3C to facilitate comparison. Protein band quantifications, normalised by eIF2α as a loading control and given relative to the mock, are provided for p-eIF2α. RT-PCR analysis of *Xbp1-u* and *Xbp1-s* mRNAs (lower panels), performed as described in Fig 2D. Immunoblots and agarose gels are representative of three biological replicates. (**C**) RT-qPCR (performed as described in Fig 2B) of *Bip*, *Chop*, *Calreticulin* and *Grp94* mRNA for three biological replicates of a timecourse of MHV infection under no-drug or IREi 60 μM/AEBSF 100 μM treatment conditions. (TIF)

**S7 Fig. Mechanistic analysis of UPR activation by SARS-CoV-2 proteins.** HEK-293T cells were treated with tunicamycin (Tn) or transfected with plasmids encoding SARS-CoV-2 S (S-HA), N (N-FLAG), ORF3 (ORF3-FLAG), ORF8 (ORF8-FLAG), pcDNA.3 as an empty vector (EV) control or mock-transfected as indicated. Cells were harvested at 24 and 36 h p.t. Western blot analysis (upper) of HERP, BiP, PERK, ATF4, p-eIF2α, eIF2α, ATF6, N-FLAG and either S-HA (**A**) or ORF3-FLAG and ORF8-FLAG (**C**). The specific p-eIF2α and ATF6-Nt bands are indicated with red asterisks. Protein band quantifications, normalised by eIF2α as a loading control and given relative to the mock, are provided for HERP, BiP, p-eIF2α and ATF6-Nt below the respective immunoblots. RT-PCR analysis of *XBP1-u* and *XBP1-s* mRNAs (lower), performed as described in Fig 2D. Immunoblots and agarose gels are representative of three biological replicates. (**B**) RT-qPCR of *BIP* and *CHOP* mRNA from two biological replicates of HEK-293T cells transfected with SARS-CoV-2 N, S or ORF8, or pcDNA.3 as an empty vector, and harvested at 24 and 36 h p.t. "h p.t." = hours post-transfection. (TIF)

**S8 Fig. UPRi treatment reverses the activation of the UPR by SARS-CoV-2 proteins. (A)** HEK-293T cells were transfected with a plasmid encoding SARS-CoV-2 S (S-HA), mock-transfected, or treated with tunicamycin (Tn). At 8 h p.t., cells were treated with UPR inhibitors (5 μM PERKi, 2 μM ISRIB, 60 μM IREi or 100 μM AEBSF) and then harvested at 24 and 36 h p.t. Western blot analysis (upper) of S-HA, HERP, BiP, PERK, ATF4, p-eIF2α and ATF6. GAPDH and eIF2α are used as loading controls. The specific p-eIF2α and ATF6-Nt bands are indicated with red asterisks. Protein band quantifications, normalised to the loading controls and given relative to the mock, are provided for HERP, BiP, p-eIF2α and ATF6-Nt below the respective immunoblots. RT-PCR analysis using primers flanking the *XBP1* splice site (middle panel), performed as described in Fig 2D. RT-qPCR (lower) of *BIP* and *CHOP* mRNA from two biological replicates of HEK-293T cells transfected and treated as described in panel A. Data are normalised as described in Fig 2B. (**B**) HEK-293T cells were transfected with a plasmid encoding SARS-CoV-2 ORF8 (ORF8-FLAG). At 8 h p.t., cells were treated with 60 μM

IREi or 100 μM AEBSF and then harvested at 24 and 36 h p.t. Western blotting (upper), RT-PCR (middle) and RT-qPCR (lower) were performed as described in panel A. Immunoblots and agarose gels are representative of three biological replicates. RT-qPCR data of SARS-CoV-2 N, S, ORF8 and empty vector in panels A and B are reproduced from S7B Fig for comparison to the treated conditions, which were performed as part of the same experiment. **(C)** RT-qPCR of *BIP* and *CHOP* mRNA from two biological replicates of HEK-293T cells transfected with SARS-CoV-2 S and ORF8 harvested at 24 and 36 h p.t. Cells were treated with 60 μM IREi and 100 μM AEBSF or a no-drug treatment control. Data are normalised as described in Fig 2B. "h p.t." = hours post-transfection.
(TIF)

**S9 Fig. Effect of the UPR inhibitors on SARS-CoV-2 infection. (A)** RT-qPCR of *BIP* and *CHOP* mRNA from three biological replicates of Vero CCL81 cells infected and treated as described in Fig 5A. Data are normalised as described in Fig 2B. Caco2 **(B)**, Vero CCL81 **(C)** or Calu3 cells **(D)** were treated with the different UPR inhibitors at the indicated concentrations for 24 h. Experiments were performed in triplicate using CellTiter-Blue Cell Viability Assay to assess metabolic capacity **(B, C** left panel and **D)** and in duplicate using trypan blue exclusion assay to assess cell proliferation and viability **(C** middle and right panels) in selected conditions. Percentages are given relative to untreated cells. **(E)** Infectivity of lentiviral particles pseudotyped with SARS-CoV-2 S protein. HEK-293T cells were transfected with ACE2 and TMPRSS2 and treated with 100 μM AEBSF between 0 and 4 h. Lentiviral particles engineered to contain a firefly luciferase reporter were pseudotyped with SARS-CoV-2 S, or vesicular stomatitis virus glycoprotein (VSV-G) as a positive control (c+, red) and empty pcDNA 3.1 vector as a negative control (c-, orange). Infectivity was measured as firefly luciferase units. Values show the mean averages of three biological replicates. Error bars represent standard errors. All *t*-tests are two-tailed and do not assume equal variance for the two populations being compared. "ns" = not significant.
(TIF)

**S10 Fig. Cytotoxicity assays of UPR inhibitors KIRA8 and Ceapin-A7.** 17 Cl-1 **(A)**, Vero CCL81 **(B)** or Calu3 cells **(C)** were treated with 10 μM KIRA8 or 15 μM Ceapin-A7 as individual treatments or in combination for 8 h **(A)** or 24 h **(B-C)**. Experiments were performed in triplicate using CellTiter-Blue Cell Viability Assay to assess metabolic activity and in duplicate using trypan blue exclusion assay to assess cell proliferation and viability. Percentages are given relative to untreated cells and doted lines represent 80% of the mock value.
(TIF)

**S11 Fig. Effect of specific inhibition of IRE1α and ATF6 pathways on MHV and SARS-CoV-2 infected cells.** Western blot analysis **(A** and **B** upper left panels) of MHV N, PERK, ATF4 and p-eIF2α from cell lysates of MHV-infected cells (MOI 5) treated with 10 μM KIRA8 or 15 μM Ceapin-A7 as individual treatments **(A)** or in combination **(B)**. The inhibitors were added to the cells immediately after the virus adsorption period and maintained in the medium until cells were harvested 8 h later. Protein band quantifications, normalised by eIF2α as a loading control and given relative to the mock, are provided for MHV N and p-eIF2α below the respective immunoblots. RT-PCR analysis of *Xbp1-u* and *Xbp1-s* mRNAs **(A** and **B** lower left panels) was performed as described in Fig 2D. RT-qPCR of *Grp94* mRNA **(A)** or MHV vRNA, *Grp94*, *BiP* and *Chop* **(B)**, performed for three biological replicates and normalised as described in Fig 2B. Immunoblots and agarose gels are representative of three biological replicates. **(C)** Plaque assays were performed with serial dilutions of the supernatant containing released virions from **(A)** and **(B)**. **(D)** Viral titres of Calu3 cells infected with SARS-CoV-

2 at MOI 5 and treated with the different UPR inhibitors a described in Fig 6. Plaque assays were performed with serial dilutions of the supernatant containing released virions harvested at 24 h p.i. In all cases, values show the mean averages of the titration of three biological replicates. Error bars represent standard errors. All *p*-values are from comparisons with the respective untreated control, with $^*p < 0.05$, $^{**} p < 0.01$, $^{***} p < 0.001$ and $^{****} p < 0.0001$.
(TIF)

**S1 Table. RiboSeq and RNASeq library composition.** Number of reads assigned to each category. Reads under 25 nt long were designated "too short".
(XLSX)

**S2 Table. Differential gene expression results. Sheets 1–4:** Ranked lists of genes which passed the thresholds of $\log_2$(fold change) greater than or equal to 1 (corresponding to a fold change of 2) and *p* value less than or equal to 0.05 after false discovery rate (FDR) adjustment for multiple testing. Sheets are as follows: TS_up, TS_down–genes which are significantly more (up) or less (down) transcribed in infected samples compared to mock, TE_up, TE_down–genes which are significantly more (up) or less (down) efficiently translated in infected samples compared to mock. Read counts for each sample, normalised by the total number of host-mRNA-mapping reads for that sample, are given in the right-most columns. **Sheets 5 and 6:** Full lists of genes which pass the threshold for inclusion in the analyses (requires ten reads mapping to this gene between all samples). TS = transcription, TE = translation efficiency.
(XLSX)

**S3 Table. Reactome pathway and GO term enrichment analysis results. Sheets 1–4:** Enriched Reactome pathways. Lists of mouse gene names of significantly differentially expressed genes (S2 Table) were used for Reactome pathway enrichment [11], in which they were converted to their human orthologues and analysed to determine which pathways are significantly over-represented. Input gene lists are indicated in the sheet name, for example 'Reactome_TS_up' shows the Reactome enrichment results generated using the 'TS_up' list from S2 Table as input. **Sheets 5–8:** Enriched GO terms. The same differentially expressed mouse gene lists were used for GO term enrichment analysis by PANTHER [114], against a background list of all the genes which passed the threshold for inclusion in that expression analysis. Column labels are as described in both Reactome and PANTHER user guides. All results with significant *p* values ($\leq 0.05$) are shown.
(XLSX)

**S4 Table. List of genes classified as translationally resistant to eIF2α phosphorylation, based on Andreev et al [18].** The first column shows the human genes classified as p-eIF2α-resistant by Andreev *et al* (excluding those from IRESite, which were not found to be p-eIF2α-resistant in their study). The list of genes used for the enrichment analysis, displayed in the second column, was generated by identifying mouse homologues of the human genes using the NCBI Homologene database [Database resources of the National Center for Biotechnology Information. *Nucleic Acids Research* **44**, (2016)]. Genes which were significantly more efficiently translated during MHV infection compared to mock infection are highlighted in bold.
(XLSX)

**S5 Table. List of oligonucleotides.**
(DOCX)

**S6 Table. Raw data analysis.**
(XLSX)

**S1 Text. Supplementary Materials and Methods.**
(PDF)

## Acknowledgments

SK, ND, GD and EB would like to acknowledge Dr Holly Shelton, Dr Isabelle Dietrich and Dr Christine Reitmayer for their supervision in the CL3 suite, and Dr Christine Tait-Burkhard and Dr Juergen Haas for the SARS-CoV-2 isolate. BGH would like to thank Prof Frank Kirchhoff for the Calu3 cells. NI would like to thank Dr James Edgar for providing SARS-CoV-2 plasmids. NI would like to thank Dr Luke Meredith and Prof Ian Goodfellow for providing pcDNA3.1-ACE2, pcDNA3.1-TMPRSS2, pMD2-VSV-G and pNL4.3-Luc plasmids for pseudotyped virions.

## Author Contributions

**Conceptualization:** Nerea Irigoyen.

**Data curation:** Georgia M. Cook, Katherine Brown, Andrew E. Firth.

**Formal analysis:** Liliana Echavarría-Consuegra, Georgia M. Cook, Idoia Busnadiego, Andrew E. Firth, Nerea Irigoyen.

**Funding acquisition:** Erica Bickerton, Benjamin G. Hale, Andrew E. Firth, Ian Brierley, Nerea Irigoyen.

**Investigation:** Liliana Echavarría-Consuegra, Georgia M. Cook, Idoia Busnadiego, Charlotte Lefèvre, Sarah Keep, Nicole Doyle, Giulia Dowgier, Nerea Irigoyen.

**Methodology:** Liliana Echavarría-Consuegra, Georgia M. Cook, Idoia Busnadiego, Katherine Brown, Stuart G. Siddell, Nerea Irigoyen.

**Project administration:** Nerea Irigoyen.

**Resources:** Katherine Brown, Stuart G. Siddell, Erica Bickerton, Benjamin G. Hale, Andrew E. Firth, Ian Brierley.

**Software:** Georgia M. Cook, Katherine Brown, Krzysztof Franaszek, Nathan A. Moore, Andrew E. Firth.

**Supervision:** Stuart G. Siddell, Erica Bickerton, Benjamin G. Hale, Andrew E. Firth, Ian Brierley, Nerea Irigoyen.

**Validation:** Liliana Echavarría-Consuegra, Idoia Busnadiego, Charlotte Lefèvre, Nerea Irigoyen.

**Visualization:** Liliana Echavarría-Consuegra, Georgia M. Cook, Nerea Irigoyen.

**Writing – original draft:** Liliana Echavarría-Consuegra, Georgia M. Cook, Ian Brierley, Nerea Irigoyen.

**Writing – review & editing:** Liliana Echavarría-Consuegra, Georgia M. Cook, Idoia Busnadiego, Charlotte Lefèvre, Sarah Keep, Katherine Brown, Stuart G. Siddell, Erica Bickerton, Benjamin G. Hale, Andrew E. Firth, Ian Brierley, Nerea Irigoyen.

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
