## [Decision Letter · Decision Letter 0]

1 Mar 2021

Dear Dr Irigoyen,

Thank you very much for submitting your manuscript "Manipulation of the unfolded protein response: a pharmacological strategy against coronavirus infection" for consideration at PLOS Pathogens. As with all papers reviewed by the journal, your manuscript was reviewed by members of the editorial board and by several independent reviewers. In light of the reviews (below this email), we would like to invite the resubmission of a significantly-revised version that takes into account the reviewers' comments.

We ask that you address the reviewers comments constructively. Please pay special attention to recurrent issues regarding data quantification and methodological clarity. 

We cannot make any decision about publication until we have seen the revised manuscript and your response to the reviewers' comments. Your revised manuscript is also likely to be sent to reviewers for further evaluation.

Sincerely,

Benhur Lee

Section Editor

PLOS Pathogens

Benhur Lee

Section Editor

PLOS Pathogens

Kasturi Haldar

Editor-in-Chief

PLOS Pathogens

orcid.org/0000-0001-5065-158X

Michael Malim

Editor-in-Chief

PLOS Pathogens

orcid.org/0000-0002-7699-2064

Reviewer's Responses to Questions

**Part I - Summary**

Reviewer #1: The manuscript by Echevarria-Consuegra et al investigates the cellular response to the infection by the murine coronavirus MHV at the transcriptional and translational level. High-throughput technologies, RNAseq and ribosome profiling, were applied, showing that the unfolded protein response (UPR) was one of the most significantly up-regulated pathways.

Coronavirus infection of cell cultures was previously shown to cause ER stress and induce the UPR in order to restore the ER homeostasis by inducing translation shutdown and increasing the ER folding capacity.

The most original aspect of the work is the study of the induction of the UPR by SARS-CoV-2 S and ORF8 proteins. In addition, different inhibitors of UPR pathways were assayed in infected cells, showing a significant reduction (1000-fold) of viral titers. These results, although preliminary, suggest that these compounds might represent potential broad-spectrum host-directed antivirals not only against coronaviruses, but also against other viruses that require the cellular UPR.

The paper provides relevant information regarding host–virus interactions at the transcriptional and translational level. However, there are some methodological concerns that should be addressed.

Reviewer #2: The authors characterize the activation of the ER unfolded protein response by murine hepatitis C (MHV) coronavirus and SARS-CoV-2 and explore branch-specific pharmacologic inhibitors of the UPR to slow virus infection. The authors start with a thorough RNA-seq and ribosome profiling-based characterization of the transcriptional and translational response of MHV infection. They identify the UPR as one of the most prominent upregulated pathways. UPR activation is well known as an adaptive response to coronavirus and other viral infections making these findings not particularly surprising or new (see ref. 28 and others), however the unbiased characterization of upregulated UPR targets, the datasets, and validations in Fig. 2 are overall valuable. The authors then explore the use of previously established pharmacologic inhibitors of individual UPR branches to inhibit MHV infection using these molecules alone and in combination. Overall, the effects on viral titers are very modest with the highest reduction (IRE1i/AEBSF) less than 100-fold (Fig. 3B) and the single inhibitors at most 10-fold. One concern is that several of the inhibitors exert cytopathic, or at least cytostatic effects (Fig. S4). In particular, the IRE1i/AEBSF combination reduces cell proliferation by ~40%, so one is left to wonder whether some of the reduction in MHV titers could simply be attributed to slower cell proliferation or cytotoxic effects. Aside from a short sentence pointing to Fig. S4, the authors do not discuss this data further, which would be important to address.

Several of the inhibitors are well validated and specific to select UPR branches (the IRE1 RNAse inhibitor STF-083010, the PERK inhibitor GSK-2606414, and the integrated stress response inhibitor ISRIB). The authors do an exhaustive job validating the inhibition of the specific UPR pathways by the compounds in the context of MHV infection, however, some data suggests cross-talk or inhibition across pathways, or incomplete inhibition (see comments below). On the other hand, AEBSF is a highly promiscuous arylsulfonyl chloride serine protease inhibitor. It prevents ATF6 activation by inhibiting site-1-protease (S1P). Besides ATF6 cleavage, S1P has other important functions, including cleavage of sterol regulatory element-binding protein (SREBP). Therefore, inhibition of AEBSF could have a variety of pleiotropic effects and the antiviral activity could be entirely unrelated to ATF6 inhibition. In addition, inhibition of other serine (or cysteine) proteases by AEBSF could be relevant during MHV or SARS-CoV-2 infections, including the potential inhibition of viral 3Cpro or PLpro proteases. The only other protease excluded as a potential off-target of AEBSF is TMPRSS2, which is required for SARS-CoV-2 entry. Considering that the authors tie the reduction in viral replication to UPR inhibition, it would especially be important to validate that the effect is directly linked to ATF6 inhibition. One easy way to address this would be to use a more selective pharmacologic ATF6 inhibitor Ceapin-A7 (see Gallagher et al., Elife, 2017 (5), e11878), which acts upstream of S1P cleavage.

Overall, the broad off-target effects of AEBSF dampen any potential therapeutic application of this inhibitor combination for coronavirus treatment.

**Part II – Major Issues: Key Experiments Required for Acceptance**

Reviewer #1: 1. RNAseq and Riboseq analysis of MHV infected cells was performed at 5 h post-infection. This time point seems to be too early to see the effect of expression of viral proteins on the cell. In fact, Fig. 2B shows at 5 hpi very low levels of Chop and Gadd34 mRNA, suggesting that the UPR was not activated yet, and other experiments in the paper were performed at 8 hpi.

2. Only SARS-CoV-2 S, N, 3a and 8 proteins were assayed to study the activation of the UPR. The rationale to exclude other viral proteins, such as E, M or accessory 6 and 7 proteins should be provided.

3. Fig S7. The activation of IRE1 pathway by S protein is clearly shown by the increase in Xbp-s RNA. However, the activation of ATF6 by S protein at 36 hpt is not that evident, since the band of ATF6-Nt is not too different from that of Mock and N. In fact, in S7B, the level of Bip expression in S-transfected cells at 36 hpt is similar to the empty vector.

4. Fig. 5B. Caco 2 cells were used to analyze the relevance of the UPR in SARS-CoV-2 infection. The lung cell line Calu 3, also used in Fig. 5C, would be a more physiological experimental model, since SARS-CoV-2 is mainly a respiratory virus.

5. Fig. 5. All the experiments evaluating the antiviral effect of UPR inhibitors should be accompanied by cytotoxicity assays to exclude the impact of cell viability on viral production.

6. Lines 394-395. A significant reduction in the expression of MHV N protein (Fig 3C) and SARS-CoV-2 S protein (Fig 5A) is observed in the presence of UPR inhibitors. As discussed, this observation might have different interpretations that should be addressed.

First, as indicated above, citotoxicity caused by the inhibitors should be excluded. Then, an indirect inhibitory effect on viral early replication should be analyzed by measuring the viral RNA synthesis at early times post-infection.

7. Lines 435-439. Actually, no significant sequence homology is observed between SARS-CoV and SARS-CoV-2 8 protein. Therefore, differences in UPR activation might have multiple causes besides the presence of VLVVL motif.

Reviewer #2: 1. The authors use different time points for characterizing the UPR activation: 5 hpi (MHV) for RNA-seq and ribosome profiling, 2.5 – 8 hpi for later characterization. But then for SARS-CoV-2, UPR activation is characterized much later (24 & 48 hpi). Could the authors explain the relevance of the timing, especially in respect to the viral replication cycles for these viruses.

2. Induction of UPR markers on Western blot should be quantified more carefully and statistics should be provided. It is unclear if the Western blots are representative, and if so, how many other replicates were analyzed. Some of the reduction in signal seems small, for example the reduction in eIF2alpha phosphorylation in response to PERKi treatment in Fig. S5A. It would be important to provide a more quantitative measure for most of the Western blot data.

3. STF-083010 only leads to a very small reduction in XBP1 splicing (Fig. S5E, Fig. 6A-B) suggesting incomplete inhibition. No direct transcriptional targets of IRE1/XBP1s are analyzed as reporters of this pathway to confirm that the inhibitor can attenuate induction. For example, ERdj4 or P58ipk could be used (see lines 214). This would be important to confirm successful inhibition of IRE1 in the context of infection.

4. When constitutively overexpressing the recombinant S, Orf8, or Orf3a proteins, one would expect some of the UPR stress to be attenuated after some time. This should especially the case for the PERK branch and eIF2alpha phosphorylation. It is surprising and not clear from the Western blot (Fig. S7) that eIF2alpha remains phosphorylated at later time points. Overall, the characterization of UPR activation in response to Orf8 and S overexpression would be much clearer with bar graphs and more careful quantification of expression or phosphorylation changes.

5. For SARS-CoV-2 infections, only the combinations of inhibitors were tested. In Fig. 5B, one striking observation is that it is mostly the combinations with AEBSF that are highly effective at reducing SARS-CoV-2 titers, raising a concern that much of the activity could be due to broad serine protease inhibition by AEBSF. What is the activity of individual inhibitors? This would be important to test. Why was such a low MOI (0.01) used for infections in Caco2 cells? Are the inhibitors still effective at higher MOI (as tested in Vero in Calu3)? Only cell metabolism (CellTiter Glo) is quantified in Calu3 and Caco2. Did the compounds similarly reduce cell proliferation as in other cells and could this contribute to the antiviral activity?

6. It seems surprising that the inhibitors are so much more effective against SARS-CoV-2 than MHV given that UPR activation for both strains is comparable. In the absence of data highlighting that the antiviral activity of the inhibitors can be directly attributed to ATF6 and/or IRE1, another likely possibility is that SARS-CoV-2 is more sensitive to broad-spectrum protease inhibition by AEBSF.

**Part III – Minor Issues: Editorial and Data Presentation Modifications**

Reviewer #1: 1. Regarding the activation of the UPR by coronavirus proteins, the paper by deDiego et al. (PLoS Pathog. 2011) showing that SARS-CoV E protein down-regulated the IRE-1 signaling pathway should be quoted (paragraph in lines 309-316).

2. The discussion should mention that the phenotypes observed in cells overexpressing SARS-CoV-2 S proteins may not reflect their physiological functions in the real infection. To obtain physiologically relevant results, experiments with recombinant viruses with deletions or modification of the target viral proteins would be required.

Reviewer #2: 1. The number of COVID-19 cases with severe diseases pathology (15%) seems high. This should be validated, and a reference provided.

2. It seems puzzling that little attenuation of eIF2alpha phosphorylation is observed. Typically, during UPR activation, phosphorylation spikes at 2-6 hrs and decreases again relatively rapidly (for example Fig. 1A – Tm lanes). GADD34 is clearly induced as early as 5 hpi, which should lead to dephosphorylation. Could the authors explain this inconsistency?

3. In Fig. 2C, it looks like polysome amounts are low in both the mock and MHV treated conditions. Shouldn’t the observed increase in 80S monosomes lead to a simultaneous decrease in polysomes?

4. For the XBP1u/s RT-PCR reactions, why is there a third high-MW PCR product? Likewise, there seems to be high degree of background splicing activity in the absence of UPR stress? Do the authors have an explanation?

5. The Western blot in Fig. 3C should be quantified. Why do some viral protein amounts increase with inhibitor treatments (for instance PERKi/ISRIB and ISRIB/IRE1i)?

6. Upregulation of UPR protein markers often lags the transcriptional induction. That could explain the lack of BiP induction in Fig. 2F and S5A. Have the authors looked at later time points post infection? It would be good to blot for other ATF6-regulated genes, for example Grp94, PDIA4, calreticulin to corroborate RT-qPCR data showing ATF6 upregulation.

7. Multiple cell models and MOI were used for SARS-CoV-2 infections, including Vero CCL81, Caco2 and Calu3 and MOI from 0.01 – 1. Was UPR activation by SARS-CoV-2 observed in Caco2 and Calu3 cells?

8. It is not surprising that the authors found it challenging to detect ATF6 cleavage by Western blot in Fig. S2. The abcam ATF6 antibody clone is likely not good choice. Better ATF6 antibodies are: Anti-ATF6alpha antibody, mouse monoclonal (37-1) (for mouse) and Anti-ATF6 � antibody, mouse monoclonal (1-7) (for human) from BioAcademia.

PLOS authors have the option to publish the peer review history of their article (what does this mean?). If published, this will include your full peer review and any attached files.

Reviewer #1: No

Reviewer #2: No
---

## [Editor Report · Decision Letter 1]

13 May 2021

Dear Dr Irigoyen,

We are pleased to inform you that your manuscript 'Manipulation of the unfolded protein response: a pharmacological strategy against coronavirus infection' has been provisionally accepted for publication in PLOS Pathogens.

We commend the authors for constructively addressing the reviewers' comments. This revised manuscript is markedly improved.  The use of more specific UPR pathway inhibitors combined with more extensive quantification, methodological clarification, and explication of underlying complexities will make this study a significant contribution to the literature.     

Best regards,

Benhur Lee

Section Editor

PLOS Pathogens

Benhur Lee

Section Editor

PLOS Pathogens

Kasturi Haldar

Editor-in-Chief

PLOS Pathogens

orcid.org/0000-0001-5065-158X

Michael Malim

Editor-in-Chief

PLOS Pathogens

orcid.org/0000-0002-7699-2064
---

## [Editor Report · Acceptance letter]

26 May 2021

Dear Dr Irigoyen,

We are delighted to inform you that your manuscript, "Manipulation of the unfolded protein response: a pharmacological strategy against coronavirus infection," has been formally accepted for publication in PLOS Pathogens.

Best regards,

Kasturi Haldar

Editor-in-Chief

PLOS Pathogens

orcid.org/0000-0001-5065-158X

Michael Malim

Editor-in-Chief

PLOS Pathogens

orcid.org/0000-0002-7699-2064